# Automated geological map deconstruction for 3D model construction using *map2loop* 1.0 and *map2model* 1.0

Mark Jessell[1], Vitaliy Ogarko[2,6], Yohan de Rose[3], Mark Lindsay[1], Ranee Joshi[1], Agnieszka Piechocka[1,4], Lachlan Grose[3], Miguel de la Varga[5], Laurent Ailleres[3], Guillaume Pirot[1]

5  [1] Mineral Exploration Cooperative Research Centre, Centre for Exploration Targeting, School of Earth Sciences, The University of Western Australia, Perth, Australia
[2] International Centre for Radio Astronomy Research, The University of Western Australia, Perth, Australia
[3] School of Earth, Atmosphere and Environment, Monash University
[4] CSIRO, Mineral Resources – Discovery, ARRC, Kensington WA, Australia
10  [5] Computational Geoscience and Reservoir Engineering, RWTH Aachen, Germany
[6] ARC Centre of Excellence for all Sky Astrophysics in 3 Dimensions (ASTRO 3D)

*Correspondence to*: Mark Jessell (mark.jessell@uwa.edu.au)

**Abstract.** At a regional scale, the best predictor for the 3D geology of the near-subsurface is often the information contained in a geological map. One challenge we face is the difficulty in reproducibly preparing input data for 3D geological models. We present two libraries (*map2loop* and *map2model*) which automatically combine the information available in digital geological maps with conceptual information, including assumptions regarding the subsurface extent of faults and plutons to provide sufficient constraints to build a prototype 3D geological model. The information stored in a map falls into three categories of geometric data: *positional data* such as the position of faults, intrusive and stratigraphic contacts; *gradient data*, such as the dips of contacts or faults and *topological data*, such as the age relationships of faults and stratigraphic units, or their spatial adjacency relationships. This automation provides significant advantages: it reduces the time to first prototype models; it clearly separates the data, concepts, and interpretations; and provides a homogenous pathway to sensitivity analysis, uncertainty quantification and Value of Information studies that require stochastic simulations, and thus the automation of the 3D modelling workflow from data extraction through to model construction. We use the example of the folded and faulted Hamersley Basin in Western Australia to demonstrate a complete workflow from data extraction to 3D modelling using two different Open Source 3D modelling engines: *GemPy* and *LoopStructural*.

## 1 Introduction

The 3D description and quantification of geometries displayed by deformed rocks has a long history (Sopwith, 1834; Argand, 1911; Ramsay, 1967; Ragan, 2009), however given the technologies available at the time, these were typically manual calculations extracted from photos or sketches. It has also long been recognised that a geological map and its legend provide more than just the distribution of lithological units but is a compendium of many different types of information (Varnes, 1974, Bonham-Carter and Broome, 1998). Burns (1988) pioneered the analysis of maps in terms of the spatial and temporal relationships stored within, and Harrap (2001) defined a legend language with the aim of consistency checking both *during* and *after* map creation, and especially when large, complex compilation maps were being created and to focus on areas where a legend contradicts map relationships. Extracting information from digital GIS maps was pioneered in the context of mineral prospectivity (Bonham Carter, 1994), and more recently to validate the maps and analyse specific structures such as stratigraphic contacts and faults, and even stratigraphic thicknesses (Fernández et al., 2005; Rauch et al., 2019; Kelka et al., 2020, Allmendinger, 2020). 3D modelling packages often have basic data ingestion schemes that can import GIS data, for example the open source package *gemsis* (https://github.com/cgre-aachen/gemgis) is an example of a system to speed up ingestion of data into the *GemPy* 3D modelling platform, which assumes that the data is already in the fundamentally correct format (e.g. contact data has already been parsed to determine the base of the unit). Since its inception, 3D geological modelling platforms have varied in their use of primary observations and geologic knowledge to constrain the 3D model geometry (Wellmann & Caumon, 2018). At one extreme the kinematic code Noddy (Jessell, 1981; Jessell & Valenta, 1986) almost exclusively uses a high-level synthesis of the understanding of structural evolution provided by the model builder to build the 3D model. Hybrid approaches that include kinematic descriptions with specific located observations are also possible (Moretti, 2008; Bigi et al., 2013; Laurent et al., 2013). In contrast, most current systems draw upon the interpolation of geological orientation and contact information to represent surfaces between observations in 3D, using direct triangulation or by interpolation of the data which can be directly observed or interpreted from geophysical data, , (Mallet, 1992; Houlding, 1994; Wu et al., 2005; Caumon et al., 2009). Approaches of this type are implemented in a range of commercial software packages (Calcagno et al., 2008; Cowan al., 2003), and more recently Open Source systems (de la Varga et al., 2019; Grose et al., 2021). In the earliest systems, the topological relationships between subsequent series, and the relative age of faults in a fault network were enforced through the construction of surfaces representing presumed structural relationships (Mallet, 2002; Caumon et al., 2004, 2009). More recently, developments have been made in methods that combine observed data and topologic and geologic knowledge in an "implicit" approach (Lajaunie et al., 1997; Aug et al., 2005; Frank et al., 2007; Caumon et al., 2013; Calcagno et al., 2008; Hillier et al. 2014; de la Varga et al., 2019; Grose et al., 2021).

The first steps in these 3D modelling workflows are time consuming, revolving around the extraction and decimation of the source data. These steps are, for the most part irreproducible: two different geologists will produce different 3D models from the same source data, and even the same geologist building the model twice would be unable to exactly reproduce the same model. In addition, the tracking of the provenance of information and decisions leading to modelling choices is effectively impossible. In this study we present the first attempts at improving that part of the 3D modelling workflow related to the transformation from map data to first model, which is one of the most time-consuming parts (hours to days) of the pre-model-building process. As discussed in this paper, this transformation is not unique but depends on the parameters used to select which features to model and the methods of combining the source datasets. This may even involve combining maps with different legends (Colman-Sadd et al., 1997), however, to date we have not addressed this issue. This study is aimed at hard-rock regional modelling scenarios which are generally data-poor compared to mines and sedimentary basins, and is part of the *Loop* project, a OneGeology consortium to build a new Open Source framework for 3D geological modelling (Ailleres et al., 2018; http://Loop3D.org). The aim of the libraries described here is to provide 3D modelling systems with a unified method for accessing legacy digital geological data, either from local files or online data servers, and to extract the maximum geological information available for use as constraints on the 3D modelling process, as well as other studies. Indeed, much of the

information extracted from the map (local stratigraphic information, the topology of fault networks, local stratigraphic offsets across faults, local formation thickness) helps in understanding the geology of the area even without building a 3D model. One might want to automate these currently manual data manipulations for many reasons, in particular for considerations of speed; reproducibility; and separation of data, concepts, and interpretations. Although the primary aim of this study was to provide information for 3D modelling workflows, some of the outputs may be useful for 2D analyses.

Jessell et al., 2014 consider four 3D Geological Modelling scenarios: Local (Mine) Scale models; Regional Scale Sedimentary Basins; Regional Scale Hard Rock Terranes and Large Scale (Crustal or Lithospheric) Models. The present work is focuses on the Regional Hard Rock Terranes scenario, where, the best predictor for the 3D geology of the subsurface is the information contained in a geological map and if available, logged well data. Unfortunately, with the exception of basin settings, drill-holes are often too shallow to provide constraints at the regional scale, and also often lack stratigraphic information (see for example the GSWA Drillhole database, http://www.dmp.wa.gov.au/geoview).

Starting from standard Geological Survey of Western Australia (GSWA) map products, and by extracting primary (e.g. stratigraphic contact location) and secondary (e.g. local formation thickness) geometric information, as well as fault and stratigraphic topological relationships, we are able to export a complete input file for two Open Source geomodelling packages (*GemPy* de la Varga et al., 2016; *LoopStructural*, Grose et al., 2020). In principle this workflow could be extended to work with other implicit modelling platforms such as EarthVision (Mayoraz et al., 1992), Geomodeller (Calcagno et al., 2008), Gocad-SKUA (Mallet, 2004) and Leapfrog (Cowan et al., 2003), although the generated input dataset may contain data that are not considered in the modelling workflow proposed by some of these packages. The idea of extracting information to feed 3D modelling algorithms directly from other data sources such as satellite data has been previously demonstrated by Caumon et al. (2013) and Wellmann et al. (2019). A parallel study building libraries for automating information extraction from drill hole data is presented by Joshi et al. (2021), so this toolset will not be discussed further here. Similarly, although geological cross-sections can be handled by similar methods to those that are described here, for simplicities sake we will not discuss them here.

In addition to the *map2model* library, *map2loop* depends on, but is being developed independently of, a number of external Open Source libraries, and in particular draws heavily on *Geopandas* (to manage vector geospatial data; https://geopandas.org/), *Rasterio* (to manage raster geospatial data; https://github.com/mapbox/rasterio), *Networkx* (to manage network graphs; https://github.com/networkx/networkx) and *Shapely* (to manage 2D computational geometry; https://github.com/Toblerity/Shapely).

## 2 Input Data

For clarity, we refer to 'inputs' as the inputs to *map2loop* and *map2model* libraries and 'augmented data' as the products of *map2loop*. The augmented data in turn form the inputs to the target 3D geological modelling engines. All temporary inputs and outputs from the related *map2model* library are wrapped within the *map2loop* library.

The information contained in a geological map falls into three categories of geometric data: *positional data* such as the position of faults, intrusive and stratigraphic contacts; *gradient data*, such as the dips of contacts or faults and finally spatial and temporal *topological data*, such as the age relationships between faults and stratigraphic units. As modellers we combine all of these direct observations with conceptual information: knowledge from near-by areas; our understanding of the tectonic history of the region, including assumptions regarding the subsurface geometry of faults and plutons, and generic geological knowledge (such as our understanding of pluton emplacement mechanisms) to provide sufficient constraints to build a 3D geological model. Often, these concepts are communicated via geological cross-sections supplied with the map, however these are typically based on limited or no additional data as they combine the conceptual ideas mentioned above with local positional and gradient information derived from the map, although they can now routinely be validated using regional geophysical

datasets such as gravity and magnetics (Spampinato et al., 2015; Martin et al., 2013). Even when we have seismic reflection data in basins, the role of conceptual biases cannot be ignored (Bond et al., 2007; Bond, 2015) In addition, the map will usually supply a stratigraphic column that provides a more direct but simplified representation of stratigraphic relationships.

In this study we draw inspiration from existing manual workflows and structural decision-making processes by developing a

suite of algorithms that allow us to automatically deconstruct a geological map to recover the necessary positional, topological and gradient data as inputs to different 3D geological modelling codes. Some of the code simply reproduces the 3D modelling packages' abilities to import different datasets, however much of it is dedicated to extracting information that is contained within the map but rarely extracted from it in a systematic fashion, as it can be rather tedious to do so, although systems such as GMDE certainly help (Allmendinger, 2020).

The libraries described here retrieve information from GIS layers or online servers, clean and decimate the data if needed, and then go through a series of data analysis steps to extract information from GIS layers stored locally or on online servers. This information includes: the local stratigraphy, the geometries of the basal contacts of units, and faults, estimates of local offsets along faults, and estimates of local formation thickness. Once these and other information have been extracted, they are output as standard formats (Graph Meta Language (GML), csv, geotif and ESRI shapefile formats) so that the target 3D modelling

systems can use them as they are.

Once the input parameters are defined, it is important to emphasise that the entire workflow is automated, so all decisions about choices of parameters are made up front (see Table 1 for a list of these parameters) and the consequences of these decisions can be directly analysed in terms of the augmented outputs of the *map2loop* code, or via the 3D models that can themselves be automatically built from these augmented outputs. Although it is a simplification, the overall workflow is shown

in Figure 2. Once the Configuration File has been generated, and the workflow control parameters defined in the *map2loop* Control Script, all further actions are fully automated, from accessing the input data, up to and including the construction of the 3D model using *LoopStructural* or *GemPy*.

In the example we present here, we use the 2016 1:500 000 Interpreted Bedrock Geology map of Western Australia and the WAROX outcrop database (GSWA, 2016) as sources of the data needed to build a first-pass model of the region around the

Rocklea Dome in the Hamersley Region of Western Australia (Fig 1). The area consists of upright refolded folds of Archean and Proterozoic stratigraphy overlying an Archean basement cut by over 50 NW-SE trending faults that form a part of the Nanjilgardy Fault System (Thorne and Trendall, 2001).

The *map2loop* library uses the *Geopandas* library to load data from several persistent formats (ESRI shapefiles, MapInfo tab files, JavaScript Object Notation (JSON) format files) and or from a Web Feature Service (WFS). Geospatial data can be in

any standard coordinate reference system (assuming a European Petroleum Survey Group (EPSG) code is supplied, http://epsg.io). These libraries are used to load and transform the input geological geometries and attributes (Table 2).

In the following subsections, which the descriptions of the six sources of input data used by *map2loop* and *map2model* (Fig. 1), are deliberately generic, as these two libraries uses a configuration file that allows the user to define which fields in the GIS layers or WFS servers contain which information. A Jupyter notebook (http://jupyter.org) helps the user to create this

HJSON format configuration file from the input layers (*Utility 1 - Config file generator.ipynb*). The minimum input data required to run *map2loop* is described in Appendix 1.

**2.1 Chronostratigraphic Polygon and Multipolygon layer**

This vector layer describes the geology polygons which have attributes defining their chronostratigraphic. Although 3D geological models can be built from purely lithostratigraphic maps, the implicit modelling schemes targeted by *map2loop*

assume some knowledge of the stratigraphy. The chronostratigraphic Polygon layer may also contain information on the surficial geology, but for more regional analysis this is either ignored by the *map2loop* library, or a map that provides interpreted bedrock geology can be used. A prototype system that accounts for thicker cover sequences is available, but not

discussed further here. The layer may contain a mixture of single Polygons, MultiPolygons (sets of Polygons with the same non-spatial attributes), and or Polygons with holes (also stored as MultiPolygons, Fig. 3). We capitalise these terms as they refer to specific *Geopandas* data objects, rather than generic geometric descriptions. Each Polygon needs to contain:

    a) a list of the ordered closed-loop x,y locations of the defining vertices,

    b) a stratigraphic code or name at a lower hierarchical level (such as formation, member), which we will refer to as 'units' (since the choice of stratigraphic resolution is up to the user, and on a map Polygons will often have different levels of stratigraphic coding),

    c) one or more higher-level stratigraphic definitions (such as group, supergroup, supersuite, province), which we will refer to as 'groups',

    d) one or more lithological descriptions that help to determine if the unit is volcanic, a sill or other types of intrusions or other types of sedimentary rocks.

    e) optionally, but importantly, the maximum and minimum estimated ages of the fine-scale stratigraphic unit.

In the case study presented here we use the 2016 1:500 000 Interpreted Bedrock Geology stratigraphic Polygons of Western Australia (GSWA, 2016). This map contains maximum and minimum estimates ages for each formation, however they may share the same ranges within a group, due to a lack of absolute geochronological constraints.

**2.2 Fault Polyline and MultiPolyline layer**

This vector layer describes the location, orientation and displacement information on mapped faults or narrow shear-zones at the surface. The layer may consist of a mixture of MultiPolylines (groups of Polylines with the same non-spatial attributes). Multipolylines are subsequently disaggregated into distinct Polylines by the *map2loop* library to allow fault length and orientation analysis to be correctly performed. Faults shorter than a user-specified length can be filtered out to reduce model complexity.

Each Polyline needs to contain:

    a) a list of the ordered open-loop of x,y locations of the defining vertices,

    b) a unique identifier so that the fault can be labelled in some way,

    c) optionally the dip and dip direction (or strike) of the fault can be stored at its midpoint.

In the case study presented here we use the 2016 1:500 000 Interpreted Bedrock Linear Features layer of Western Australia (GSWA, 2016), filtered by *map2loop* to extract the faults.

**2.3 Fold axial trace Polyline layer**

This vector layer describes the location and polarity (anticline vs syncline) information on mapped fold axial traces, defined by the intersection of the fold axial surface and the surface of the Earth. The layer may consist of a mixture of Polylines and MultiPolylines (groups of Polylines with the same non-spatial attributes).

Each Polyline needs to contain:

    a) a list of the ordered open-loop of x,y locations of the defining vertices,

    b) a unique identifier so that the fold axial trace can be labelled in some way,

    c) the polarity of the fold axial trace (syncline, synform, anticline or antiform).

In the case study presented here we use the 2016 1:500 000 Interpreted Bedrock Interpreted Bedrock Linear Features layer of Western Australia (GSWA, 2016), filtered by *map2loop* to extract the fold axial traces.

## 2.4 Bedding orientation point layer

This vector layer describes the local orientation of bedding, and is often missing from map packages, but can be found in the separate databases, or original field notebooks. It could also be estimated by photointerpretation and/or three-point analysis. The layer may consist of Points.

Each Point needs to contain:

    a) a single x,y location of the defining Point,

    b) dip information,

    c) dip direction, or strike information, which we will refer to as 'azimuth' to avoid confusion,

    d) the polarity of the bedding (upright or overturned).

In the case study presented here we use the 2016 WAROX outcrop database (GSWA, 2016).

## 2.5 Reference Stratigraphy

Some countries have developed national-level stratigraphic databases (such as the Australian Stratigraphic Units Database, ASUD, Geoscience Australia and Australian Stratigraphic Commission, 2017; https://asud.ga.gov.au/) that allow access to detailed stratigraphic information at the formation-level and above. The max-min ages for individual Polygons mentioned in Section 2.1 would typically be derived from such a database. This national-level stratigraphic information is typically non-spatial, however assuming that the mapped chronostratigraphic Polygons share the same coding as the national database, we can use this to augment the stratigraphic relationships (such as 'A overlies B') once the topological analysis has been carried out by *map2model*, which in turn help to define the local stratigraphy in the map area. The *map2loop* library currently uses a condensed extract from the ASUD database that defines neighbouring stratigraphic relationships as pairs (A overlies B) to refine the local stratigraphy (Fig. 1b).

## 2.6 Digital terrain model

This grid layer, usually derived from the SRTM (Shuttle Radar Topography Mission; Farr et al., 2007) or GDEM (Aster Global Digital Elevation Map; NASA/JPL, 2009) datasets, or a fusion of both, provides a uniform coverage of surface topography measurements over most of the continents. The *map2loop* library uses the Geoscience Australia server for 90m coverage in Australia (Geoscience Australia, 2016), the 1km global coverage offered by the Pacific Islands Ocean Observing System (https://pae-paha.pacioos.hawaii.edu/thredds/dem.html?dataset=srtm30plus_v11_land) server for coverage outside Australia, although there are a number of such servers now available, and the data is directly downloaded for the region of interest during the processing workflow. Local on-disk rasters of DTMs in geotif format may also be used.

In the case study presented here (Fig. 1c) we use the 90m version served by Geoscience Australia (Geoscience Australia, 2016).

## 2.7 Validation of Input Data

Once the sources of data are defined, an automated initial verification of the data is performed to assure that the different information needed to perform the calculations is present. First it clips the data to the region of interest and then these new layers are checked to ensure that there is sufficient bedding data, as the algorithms we use require at least three orientations to interpolate a complete bedding orientation field. Then it checks to see if the geology Polygon file has any data in it. Empty layers can arise because of data path or projection errors, so there is no point continuing the calculations if there is no data and the program stops with an error statement. We also verify that each layer has all the fields described in the Configuration file, again if required fields are missing, the program stops. Warnings will be issued if empty values are found for required fields,

or optional fields are missing, in which case default values will be provided but will not stop program execution. Some data validations take place subsequently during calculations themselves, as they depend on an analysis of the values of features, or secondary calculations as described below.

## 3 Methodology

The *map2loop* and *map2model* libraries combine the inputs described in Section 2 in different combinations to produce a series of augmented outputs as *csv, geotif* and *gml* format files that can be used directly by the target 3D geological modelling systems, or as sources of analysis for 2D studies. *map2model* performs a spatial and temporal topological analysis of the geological map, and *map2loop* further refines this analysis by including information from non-map sources, such as stratigraphic databases, acts as a wrapper for *map2model,* and performs all other calculations.

This section outlines the high-level logic of how the different inputs are combined to produce information needed by the target 3DGM systems. As with the inputs to *map2loop*, the outputs are grouped by type: *positional*, *gradient*, and *topological* outputs. The specific positional, gradient and topological outputs are in most cases calculated by combinations of the positional, gradient and topological inputs, and so the ordering below does not in general reflect the order in which these augmented data are produced by the *map2loop* library, and reference is made to data calculated in later sections. Ordering the sections by order of calculation results would be useful to get an understanding of the specific data flow (Fig. 4), but also produces a rather confusing back and forth in text form as some data is incrementally modified as the workflow progresses. Example pseudocode for key calculations is included in Appendix 2.

In the following sub-sections, we provide an overview of the different steps that the code automatically undertakes to extract augmented data from the input files. A summary of the specific outputs used by the 3D modelling engines used in this study is provided in Table 3.

### 3.1 Positional Outputs

The first class of modelling constraints derived by the *map2loop* algorithms provide positional data. Positional outputs refer to information that defines the *x,y,z* location of a feature, including the position of faults, intrusive and stratigraphic contacts. In this section we describe the combinations of data used to create these augmented data.

### 3.1.1 DTM

The online Digital Terrain Model (DTM) servers described in Section 2.6 either provide the information at a fixed x,y spatial resolution, or allow the client to subsample the data. For regional geological models a high-resolution topography model is usually not needed as the spatial resolution of 3D models is generally larger than the 30m available from SRTM data, so a 90m or even 1km DTM is often sufficient for our needs. The *map2loop* library imports a subset of the global or national DTM, which are usually provided using a WGS84 projection. This is then reprojected using the *Rasterio* library to a meter or other non-degree based projection system. This distance preserving coordinate system is appropriate for use by modelling packages that produce Cartesian models where the x,y and z coordinates use the same length units. The reprojected transformed DTM is stored as a *geotif* format file. Code is in development that will allow local geotif format DTM sources to be accessed.

### 3.1.2 Basal contacts

The *map2loop* library currently uses the convention that stratigraphic contacts are labelled by the overlying unit in the stratigraphy, so that the contacts represent the bases of units, which we will refer to as basal contacts. Basal and intrusive contacts are calculated using the intersection of neighbouring Chronostratigraphic Polygons (Section 2.1). At the moment sill-like intrusive contacts are ignored, as they do not follow either massive pluton-like geometries or strict stratigraphic

relationships, but are the current subject of further study. Although stratigraphic lenses will be processed by *map2loop*, the 3D modelling packages we currently link to are unable to deal with these features except by inserting unconformities at the top of each lens, and this remains an open area for future studies. In order to determine the label of the resulting Polyline, we analyse the stratigraphic relationship between two Polygons using the information from the local stratigraphy calculated by the

275 *map2model* library ( Section 3.3.1):

    a) if the two units are both volcano-sedimentary units, we label the basal contact with the unit name of the younger unit,

    b) if one of the units is intrusive (not a sill) and the other has a volcano-sedimentary origin, we assign the intrusive unit name if the intrusion is younger than the volcano-sedimentary unit, or the volcano-sedimentary unit if the intrusion is older,

c) if both units are intrusive (not sills) we assign the contact name to the younger unit.

    d) If one or both of the units is a sill, we ignore the contact completely.

The x,y coordinates come from the intersection Polylines, and can be decimated by taking every $n^{th}$ node, the z value comes from the DTM. Outputs from *map2loop* consist of:

    a) a series of x,y,z points,

b) unique stratigraphic name for each Polyline, and

    c) for each point the polarity of the contact (relative direction of younging and dip direction, a value of 1 means they are in the same direction and hence the bedding is the right way up, for overturned beds the value is 0)

### 3.1.3 Fault position and dimensions

Processing of fault geometries consists of extracting the x,y location of nodes from the fault Polylines (Section 2.2), combining

with the DTM to get z, and calculating the distance between fault tips to define overall fault dimensions. A minimum fault length threshold can be applied so that very short fault segments, which will have little impact on the model, can be ignored. A decimation factor that only stores every $n^{th}$ node value can also be applied. If needed, prior to *map2loop* processing, we use *FracG* (Kelka et al., 2020) to recombine fault segments based on the coincidence of fault tip locations and similar fault trace orientations.

Outputs from *map2loop* consist of:

    a) a series of x,y,z points

    b) a unique code that can be used to create a name for each Polyline, and

    c) for each fault Polyline the dip, azimuth and length of the fault

### 3.1.4 Fold axial trace position and dimensions

Processing of fold axial trace geometries consists essentially of extracting the x,y location of nodes from fold Polylines (Section 2.3), combining with the DTM to get z. Fold polarity (anticline/syncline) is recovered and stored. A decimation factor that only stores every $n^{th}$ node can be applied. Outputs from *map2loop* consist of:

    a) a series of x,y,z points

    b) unique fold axial trace name for each Polyline, and

c) for each fold axial trace Polyline the polarity of the fold

### 3.1.5 Local unit thickness

The local apparent thickness of units is calculated by finding the intersection of a line normal to the local tangent of a stratigraphic contact and the next stratigraphic contact (Fig. 5). Based on the stratigraphic relationship there are three possibilities:

a) if the next contact is the stratigraphically adjacent and higher contact, the distance is calculated ($T_a$) and stored as a local apparent thickness measurement.

b) if the next contact is stratigraphically higher, but not the stratigraphically adjacent, the distance is calculated and stored as the minimum apparent thickness ($T_m$),

c) otherwise no calculation is made.

True actual and minimum thicknesses can then be calculated from the apparent actual and minimum thicknesses as:

$$T_t = T_a \sin(\theta) \qquad\qquad 1$$

where $T_t$ is the true dip, $T_a$ is the apparent dip and $\theta$ is the dip of the bedding relative to the land surface (Fig. 5, Section 2.3.2). As these calculations can potentially be made for each node of a stratigraphic contact, we often end up with multiple estimates per unit, for which we can calculate the aggregated information as follows:

a) if we have true actual thicknesses for a unit, we store the median and standard deviation of thicknesses, and use the median of the actual thicknesses to calculate the local normalised thickness for each calculated node.

b) if we only have minimum thicknesses, we store the median and standard deviation of the minimum thicknesses and use the median of the normalised thicknesses to calculate the local normalised thickness for each calculated node.

c) if we have neither actual nor minimum thicknesses, if needed we use the median of the medians of thicknesses of all units as a rough estimate of the thickness, and no normalisation is possible.

Outputs from *map2loop* consist of:

a) a series of x,y,z points

b) apparent, actual/minimum, normalised actual/minimum thicknesses for each node and error estimates where appropriate

c) table of summary thicknesses for all units

### 3.1.6 Local fault displacement

We have implemented three distinct methods of estimating the displacement across faults, depending on data availability. The most complete analysis of fault displacements is based on identifying equivalent stratigraphic contacts across a fault and measuring their apparent offset (Fig 6a $D_a$), assuming that these are not growth faults. If we combine this with the local interpolated estimates of dip/azimuth for the whole map (Section 3.2.4), and we know the orientation of the slip vector, we can calculate the true fault offset (Fig. 5a). Unfortunately, slip vectors are often hard to measure in the field and rarely recorded in geological maps. Given this, we can make an arbitrary assumption that the slip vector is down-dip ($F_t$), and then calculate the displacement based on the dip of the bedding, and the dot product of the contact and fault trace normal as:

$$D_t = D_a \tan(\theta\ C_n \bullet F_n) \qquad\qquad 2$$

where $D_t$ is the true displacement, $D_a$ is the apparent displacement, $C_n$ is the 2D contact normal, $F_n$ is the 2D fault normal and $\theta$ is the dip of the bedding. Since these are local estimates, we can have multiple estimates along the same fault, in which case even these poorly constrained displacement estimates are of interest, as the relative displacement pattern along the fault can still be determined. Where these displacement calculations can be made, we can also determine the local downthrown block by comparing the sense of displacement (dextral or sinistral) with the dip of the strata (Fig. 5h). Specifically, the downthrown direction is given by considering the cross product of the fault tangent, the contact normal and the sign of the relative offset as follows:

$$W = (F_t \times C_n)\,\text{sgn}(D_s) \qquad\qquad 3$$

Where $W$ is the downthrow direction, $F_t$ is the fault tangent, $C_n$ is the contact normal and sgn$(D_s)$ is the sign of the apparent displacement sense (positive is dextral). If $W$ is negative, the downthrown direction is defined by the normal to the fault trace with a right hand rule, and if the result is positive, by the opposite direction. The ability to match equivalent stratigraphic contacts across a fault depends on the type of geology, the scale of the project and the detail of the mapping.

A second level of displacement estimates can be made by comparing the stratigraphic offset across the fault, so if we have a stratigraphy going from older to younger of C-B-A and a fault locally separates unit A and unit C, then we can assume the offset has to be at least the thickness of units B, so if we have estimates of unit thickness (see Section 3.1.5) then we can estimate minimum offset (Fig. 5b). If, for the same stratigraphy, the fault offsets the same unit A-A, or stratigraphically adjacent units A-B, the conservative estimate of minimum displacement would be zero.

Finally if we do not have unit thicknesses available, we can always simply record the stratigraphic offset in terms of number of units (Fig. 5b), so in the original example above, an A-C relationship across a fault can be recorded as a stratigraphic offset of 2. The last two methods are not currently used in the automated workflow to determine fault offset; however, they do provide insights into which faults are the most important in a region.

## 3.2 Gradient outputs

The second class of modelling constraints derived by the *map2loop* algorithms provide gradient data. Gradient data in this context refers to information that defines the local orientation of a feature, such as the dips of stratigraphic contacts or faults. In this section we describe the combinations of data used to create these augmented data.

### 3.2.1 Bedding orientations

The orientation data produced by the *map2loop* library is derived from a combination of gradient and positional sources, specifically the Bedding orientation point layer (x, y, dip, azimuth, polarity; Section 2.4), the DTM (z; Section 2.6) and the Chronostratigraphic Polygon layer (unit; Section 2.1). A filter is applied to remove observations where the dip is zero, as our experience has shown that this usually reflects a measurement where the dip was unknown, rather than a true dip of zero. Optionally, the number of points can be decimated based on taking every $n^{th}$ point from the layer. More sophisticated decimation procedures, such as those described in Carmichael and Ailleres (2016), for orientation data are the subject of current work. Internally the code uses a dip direction convention so if strike data are provided, we convert these to dip direction before calculation.

Secondary gradient information can be assigned along all the stratigraphic and intrusive contacts based on a series of simple assumptions:

a) the dip direction of all dips is assumed to be normal to the local tangent of the contact and are defined as zero at North and positive clockwise.

b) the dip can either be uniformly defined, or for the case of stratigraphic contacts, based on interpolated dips (see Section 2.2.4).

c) the azimuth of intrusive contacts for dome- or saucer-shaped bodies can be arbitrarily be selected by choosing the polarity of the dips and the azimuth (domes have outward dips and inverse polarity, saucers have inward dips and normal polarity).

### 3.2.2 Fold orientation

If fold axial traces are available, and in areas with otherwise sparse bedding information, it can be useful to seed the model with extra orientation information that guides the anticline-syncline geometries.

Outputs from *map2loop* consist of, for each fold:

a) x,y,z positions

b) a series of dip/azimuth pairs offset each side of the fold axial trace

c) stratigraphic unit for each position

### 3.2.3 Fault orientation

If fault orientation data is available, either as numeric dip/azimuth (e.g. dip value: 75, azimuth value: 055) or in text form (e.g. dip value such as 'Shallow, Medium, Steep, Vertical', azimuth value such as 'Northeast') then this is recovered and stored, otherwise the fault dip orientation is calculated from the fault tips, and the dip is set to a fixed value or is allowed to vary randomly between upper and lower limits. In the absence of other supporting information the qualitative dip information assumes equally spaced dips between the shallowest and steepest term, and assumes that the shallowest term is not horizontal, so in the example above we would get 'Shallow'=22.5, 'Medium'=45, 'Steep'=67.5 and 'Vertical'=90.

Outputs from *map2loop* consist of, for each fault (Fig. 7b):

a) x,y,z positions of the end-points and mid-point of the fault

b) a dip/azimuth pair for each location

### 3.2.4 Interpolated orientation field

It became apparent during the development of this library that obtaining an estimate of the dip from bedding everywhere in the map area was a necessary precursor to calculating important information such as unit thickness (Section 3.1.5), fault offset (Section 3.1.6), as well as the dips of contacts at arbitrary locations. In an attempt to retain more geological control over the sub-surface geometries, de Kemp, (1998), used polynomial and hybrid B-spline interpolation techniques to extrapolate geological structure. All more recent 3D geological modelling packages involve generalised interpolants of one form or another (Wellmann and Caumon, 2018; and see Grose et al. (2020) for a discussion of the strengths and weaknesses of the different interpolants). At the scale of the map, we observe that local bedding azimuth measurements are often relatively poor estimators of the map-scale orientation field. This occurs because the point observations record second-order structures, such as parasitic folds. In order to avoid these issues we have instead chosen to use the primary orientation data only for dip magnitudes, for which we have no alternative, and use the azimuth of stratigraphic contacts as the best estimator of the regional azimuth field. To this end we calculate a regular dip field using a multiquadratic Radial Basis Function (RBF) of the primary orientation 3D direction cosines using the *scipy* library, and separately use an RBF to interpolate the 2D contact azimuth direction cosines ($l_c$, $m_c$, Fig. 7d). Each set of orientations from structurally coherent 'super-groups' (see Section 2.4) are interpolated separately. For each super-group, we then combine these into a single direction cosine ($l_o$, $m_o$, $n_o$ i.e. the direction cosines of the interpolated bedding orientations) taking the $n_o$ value from the interpolated 3D direction cosines and the $l_c m_c$ terms from the 2D direction cosines and normalising so that the vector has a length of 1. This gridded field is then available for the thickness and offset values as discussed above, but could conceivably be used with appropriate caution as additional estimates of orientation in parts of the model where no direct observations are available, or for cross-validation with known values.

### 3.3 Topological outputs

The third class of modelling constraints derived by the *map2model* algorithms provide the spatial and temporal topology of the map layers. Specifically, it creates network diagrams showing the stratigraphic relationships between units in the region of interest (Burns, 1988; Perrin and Rainaud, 2013; Thiele et al., 2016), network diagrams of the relationships between faults, and relationship tables showing whether a particular fault cuts a unit or group.

### 3.3.1 Local stratigraphy

The spatial and temporal relationships integrated into geological maps provide a key constraint for 3D geological modelling (Harrap, 2001; Perrin and Rainaud, 2013). At the scale of a map sheet, state/province or country stratigraphic legends are necessarily simplified models of the complex range of stratigraphic relationships. Since our aim is to build a model for an arbitrary geographic region, we need to be able to extract the local stratigraphic relationships rather than just relying on the high-level summaries. The *map2loop* library uses the *map2model* C++ library to extract local stratigraphic, structural and

intrusive relationships from a geological map. *map2model* uses two of the layers sourced by *map2loop*: namely the chronostratigraphic Polygon layer (Section 2.1), the fault Polyline layer (Section 2.2).

Shared contacts between Polygons defining units, calculated by an intersection calculation that results in a Polyline, are labelled as either intrusive, stratigraphic or faulted based on the nature of the units either side of the contact, and the presence or absence of a spatially coincident fault Polyline (Fig. 6). The logic is as follows:

a) if a Polyline between units coincides spatially with a fault Polyline, the Polyline is labelled as a fault contact

b) if a Polyline is between one intrusive unit and a volcano-sedimentary unit, the Polyline is labelled intrusive if the intrusive unit is younger than the other unit, or stratigraphic if it is older.

c) if the Polyline is between two intrusive units, the Polyline is labelled as intrusive.

d)  Otherwise, the Polyline is labelled as stratigraphic.

The relative age of each unit is determined from the min/max ages supplied for each unit in the map, and if these are not available, or they have the same age, or age range, then no age relationship is assigned. The primary outputs from map2model are a series of network graphs in Graph Meta Language formal (GML) that can be visualised by the free but not Open Source *yEd* package ((https://www.yworks.com/products/yed) or the Open Source *Gephi* package (https://gephi.org/). The *map2model* code provides graphs of all igneous, fault and stratigraphic contacts, and the stratigraphic relationship graph underpins the

definition of local stratigraphy in the *map2loop* system.

As not all maps provide max/min age information, *map2loop* can optionally update the stratigraphic ordering by using a national or regional reference stratigraphic database (Section 2.5). Depending on the structure of the database, an age-sorted ordering of all units in the database, or pairwise stratigraphic relationships such as 'unit A overlies unit B', have be used to refine the ordering extracted from the map. Even after these progressive refinements, ambiguities in relative age of units

usually remain. At the moment *map2loop* arbitrarily choses one of the distinct stratigraphic orderings as the basis for its calculations, but clearly this is an important source of uncertainty that could be used stochastically to explore stratigraphic uncertainty.

We can reduce the uncertainty in the stratigraphic ordering that comes from lack of information in the map as to relative ages, or ambiguous relative map age relationships, by considering one higher level of stratigraphy, which we will call 'groups' but

could be any higher rank of classification. This reduces the uncertainty as typically the uncertainty in relative ages between groups is smaller than the relative ages of any two units if we ignore their group relationships.

Since *map2loop* is primarily aimed at implicit modelling schemes, there is a considerable advantage in reducing the number of stratigraphic groups that have to be interpolated separately, since the more orientation data we have for a structurally coherent set of units the better the interpolation. To this end we use the *mplstereonet* Python library to compare each group's

best-fit girdle to bedding orientation data, so that if their respective girdle orientations within a user-defined value, they can be considered to be part of the same 'super-group'.

The outputs of *map2loop* are a stratigraphic table (csv format) defining a distinct ordering of units and groups, plus a table of which groups form super-groups to be co-interpolated.

### 3.3.2 Fault-fault relationships

The intersection relationships between pairs of faults are calculated by map2model by analysing which faults terminate on another fault. This is assumed to represent an age relationship, with the fault that terminates assumed to be the older fault. The *map2loop* library converts this information into a table of binary relationships: Fault X truncates/has no relationship to Fault Y that are then compiled into a set of graphs of fault-fault relationships.

### 3.3.3 Fault-stratigraphy relationships

The intersection relationships between stratigraphic units and groups are calculated by the *map2model* library by analysing which geological Polygons have sections which are spatially coincident with faults. These are then converted by the *map2loop* library into two tables of the binary stratigraphic relationships unit/group A is cut by/is not cut by fault X.

### 3.4 Validation of Augmented Data

Once the augmented data types have been calculated by *map2loop* and *map2model*, a final validation of the data is
automatically performed so that there are no 'orphan' data, for example orientation data for units that will not be modelled, and a unit in the stratigraphy for which we have no contacts or orientations. Although this can obviously happen in nature, current modelling systems struggle with this concept, so we need to ensure that the model will actually build by removing unresolvable data.

### 3.5 3D Modelling using *map2loop*/*map2model* augmented outputs

The two Open Source modelling packages we have targeted use overlapping source of information but distinct data formats to perform their modelling (Table 2). Some of the augmented data produced by the library are not (yet) explicitly required by any of the packages but are useful datasets for contextual regional analysis and can provide some guidance for studies un-related to 3D modelling. A partner project led by the Geological Survey of Canada is developing a Knowledge Manager to support higher level information as a geoscience ontology to provide conceptual frameworks for modelling, aggregated petrophysical
data and other basic knowledge of relevance to 3D modelling workflows (Brodaric et al., 2009; Ma and Fox, 2013).

The outputs of *map2loop* and *map2model* described above provide all of the information required to build 3D geological models, in *GemPy* (de la Varga et al, 2019) and *LoopStructural* (Grose et al., 2020).

The ability to generate all necessary input data for a geological model from set of source layers in a matter of minutes demonstrates the potential for this approach to reduce the entry barrier for geologists who wish to make 3D models as part of
their exploration or research programs.

## 5. Results

The results of the first stage of the automated workflow controlled by *map2loop* and including the *map2model* libraries are a set of augmented outputs that are both useful in their own right in terms of their ability to produce unbiased analyses of the map data, and as inputs the 3D modelling packages. A summary of all the files used by the 3D modelling engines generated
by *map2loop* and *map2model*, together with file types, is given in Table 4.

### 5.1 Results of positional calculations

The positional information extracted from the various input data include:

 a) Basal contacts of stratigraphic units (Fig. 7a), optionally decimated. Black lines show the original Polygon boundaries, and the coloured circles show the location of the base of the stratigraphic unit. Lines with no basal contacts
are sills that are not yet handled by the code, or the modelling engines

b) Fault traces, colours randomly assigned to each fault, only faults longer than a defined length, in this case 5km, are processed (Fig. 7b), optionally decimated. Some faults as mapped (near 56000, 7496000) were ignored because they formed closed loops, or were mapped with acute angles, which the modelling engines were not able to deal with properly, and are in any case unlikely to be correctly drafted in this map.

c) Fold axial traces (Fig. 7c), optionally decimated.

d) Local unit thicknesses, as apparent, true, and normalised thicknesses (each true thickness estimate divided by the median value for each unit) (Fig. 7d). In areas with sills, the code does not attempt to calculate thicknesses.

e) Fault offset, both apparent and inferred true displacement assuming down-dip displacement (Fig. 7e).

f) Fault offset derived from minimum stratigraphic offset (Fig. 7f).

g) Stratigraphic fault offset (Fig. 7f).

h) Fault downthrown block direction (Fig. 7g).

## 5.2 Results of gradient calculations

The gradient information extracted from the various input data include:

a) Bedding orientations near fold axial traces (Fig. 8a).

b) Fault orientations (Fig. 8a), optionally decimated. Fault mid-points are shown here, but the same values are also placed at each fault tip.

c) Interpolated orientation data, calculated as interpolated lc,mc , inset of part of NW area of map (Fig. 8b).

d) Interpolated contact tangents, calculated as interpolated lo,mo,no direction cosines, inset of part of NW area (Fig. 8c).

e) Combined information from interpolated dips and interpolated contacts, inset of part of NW area (Fig. 8d).

## 5.3 Results of topological calculations

The gradient information extracted from the various input data include:

a) Stratigraphic ages relationships extracted from map and ASUD. Arrows point to older unit. Thickness of arrows is proportional to contact length (Fig. 9a).

b) Fault-intersection relationship graph (Fig. 9b).

c) Subset of fault-unit truncation relationships, the green cells show stratigraphic units that are cut by faults, the yellows cells are not cut by faults (Fig. 9c).

## 5.4 Results of 3D model calculations

Once the automated data extraction has been completed the augmented data are passed to the 3D modelling engines to

automatically build the 3D geological model (Fig. 10). Note that two packages use different subsets of the available data, as well as different interpolation algorithms, and hence should not be expected to produce identical results. *GemPy* calculates limited-extent faults but currently displays them as extending across the model area. In both cases a first-pass 3D model that respects the major geological observations is produced.

## 6. Discussion

The example map and associated data used in this paper took just over 3 minutes to deconstruct with *map2loop* and a further 4-15 minutes to build with the three target modelling engines, running on a laptop computer with 32 GB of RAM and 4 i7 Intel Cores running at 1.8 GHz. The time taken to deconstruct a map depends on the number of features to be processed (polygons

+ polylines + points), with the slowest part of the calculation being the extraction of true fault displacements. The time for model construction increases systematically with the increase in resolution of the interpolation and isosurfacing calculations.

There are currently no other codes that we are aware of that perform the same automated data extraction workflows presented here, aimed at building regional 3D geological models, so questions of external code benchmarking are not possible, however we have run a comparative experiment where one of the authors (MJ) extracted the information needed to provide the inputs for LoopStructural from the raw data sources and the timing results are shown in Table 5, and the time taken to extract the data manually (over 4 ½ hours) does not compare favourably with the automated workflow (3 minutes). For a one-off map we need

to add around 20 minutes to the automated calculation time to set up the configuration file, but for any additional maps from the same map series, for which we can use the same configuration file, the start-up time is of the order of minutes.

Although the improved speed of data extraction is an advantage, the principal motivation for this study was to develop a system where the complete 3D modelling workflow, including data extraction, could be automated. This is crucial for Sensitivity Analysis, Uncertainty Quantification and Value of Information studies since all these approaches depend on our ability to

perform stochastic simulations of the whole 3D modelling workflow, which is not possible if the first manual steps remain unquantified and subject to modeller bias.

The choices made by the *map2loop* and *map2model* code are inspired by the thought processes of geologists when manually building a 3D geological model from the same data. There are many small or large decisions and assumptions that are made when developing the model, and the discussion below highlights some of the areas where further work needs to be done to

reproduce the manual workflow. In this paper we have used an example from Western Australia, however similar examples for the Northern Territories, New South Wales, Victoria, Queensland, Tasmania and South Australia can be run using the *map2loop* library using copies of data stored on the Loop WFS server (https://geo.loop-gis.org/geoserver/web).

## 6.1 Improvements to calculations

The aim of this study was to build an end-to-end workflow from raw map 'data' to a 3D model, which we hope to build upon

by refining the different steps as discussed below.

### 6.1.1 Choice of data

The code as it stands provides limited filtering of the data via decimation and the use of a fault length filter for faults. There are many different reasons for building 3D geological models, and each reason may support a different selection of the available data to ensure critical elements in the 3D model are preserved. In the case of faults, it may be, for example the fault

network itself which is important, either as barriers or pathways of fluid flow, or it may be the geometric consequences of faulting that are important, for example when the goal is to provide prior petrophysical models for geophysical inversions. Apart from fault length, these choices need currently to be made by deleting data at the source, however a future implementation of 'intelligent filtering' that made clear the reasons for data selection would remove the hidden biases from these choices.

### 575 6.1.2 Calculation of unit thickness

The calculation of local unit thickness (Section 3.1.5) depends on the local estimate of apparent unit thickness, which is reasonably robust, and has been validated by comparison with manual measurements, but also on the local estimate of the dip of the stratigraphy. This dip estimate comes from the application of the *scipy* Radial Basis Function interpolation library, and in particular the multiquadratic radial basis function, which can be supplemented by a smoothing term. Other radial basis

functions such as Gaussian and inverse are available, as well as other schemes such as Inverse Distance Weighting and co-kriging, which all offer tuneable algorithms for estimating the local orientation field. We chose the multiquadratic RBF simply because our experience showed that, for the types of geology that we started working on, it produced 'reasonable' results. It is

likely that different geological scenarios may require optimised interpolation schemes (Jessell et al, 2014) as there is no unique solution to this problem.

**6.1.3 Calculation of fault offset**

The calculation of local fault offset also relies on the interpolated dip field, so the same remarks regarding geologically appropriate interpolators stated in the previous section apply. If we compare the local displacements along a fault, then we also must assume that the unit thickness is the same on both sides of the fault, but at least in general this can be tested directly. In addition, to properly estimate fault displacement, which we have validated by manual measurements, we need to know the fault displacement vector. One solution, not yet implemented, would be to calculate the relative displacement of lines of intersection of the same dipping stratigraphic units across fold axes either side of a fault.

**6.1.4 Calculation of super-groups**

The definition of super-groups for co-interpolation of bedding data is performed by comparing the orientation of best-fit girdles. This has a number of flaws. Firstly, disharmonic fold sequences may have the same orientation spread, but different wavelengths and thus should not be interpolated together. Secondly, if a particular group is undeformed, or lies on one limb of a fold, there may not be a well-developed girdle. A more robust analysis of fold structural information, which includes analysis of representative fold profiles, as described by Grose et al., 2019, would not only allow us to better identify coherent structural domains, but would also provide the information needed to use the more sophisticated modelling schemes described in their work.

**6.1.5 Choice of stratigraphic ordering**

As described in section 3.3.1, the stratigraphic ordering of units is derived from a combination of local observations drawn from the geology Polygon and fault Polyline layers, and a regional or national reference stratigraphy. This process does not generally lead to unique stratigraphic orderings, and at present we simply take the first sorted result from a sometimes-long list of alternatives. A second unknown is the nature of the contact between different groups. We use the idea of super-groups to cluster structurally coherent domains, but we do not currently have a good solution to estimate the nature of stratigraphic discontinuities between structurally incoherent domains. The modelling systems we target allow for onlap and erode relationships, and Thiele et al. (2016) suggested the topological analysis of units to identify unique relationship characteristics between groups as a possible way forward, but this remains to be tested. Recent attempts at the automatic extraction of stratigraphic information from 3D seismic data (Bugge et al., 2019) could also constrain our systems in basin settings.

**6.1.6 Analysis of fault-fault and fault-unit topology**

The assumption that a fault or unit that truncates against another fault represents an age relationship is reasonable, but exceptions obviously exists in reactivated faults and growth faults. At the present time if a cycle in fault age relationships is discovered: Fault A cuts Fault B; Fault B cuts Fault C; Fault C cuts Fault A, one of the age relationships is removed arbitrarily. A better approach may be to look at displacement, length, or some other characteristic such as stratigraphic offset to make that decision. A further test may be the centrality of a fault, for which there are several methods (Freeman, 1977), for example related to how many other faults are truncated by a specific fault. These fault-fault and fault-unit age relationships could provide further constraints on the overall stratigraphic ordering of units, and of the structural history of a region that would be valuable inputs to time-aware modelling systems such as *LoopStructural*.

## 6.2 Limitations in resulting 3D models

Given the complexity of the task, and the limitations and somewhat arbitrary nature of some of the choices described above, it is perhaps surprising that we ever get a good 3D model out of the system. Conversely there are a number of other reasons why having deconstructed a map, we do not end up with a 3D model that meets our expectations or needs. When running the code over different types of geology, we need to distinguish between two types of results: firstly, has the code correctly and completely extracted the available data; and secondly, is this data sufficient to build a 3D geological model. Our experience

from different geological terranes, including deformed basins including the Hamersley Basin, the Yilgarn Craton Granite-Greenstone Terranes and the igneous complexes in the South-West Terrane, all in Western Australia, is that the code provides the data we would expect as geologists, but that in more complexly deformed terranes such as Granite-Greenstone belts, the 3D models do not live up to our mental images. Typically, the 3D fault networks look reasonable, but the stratigraphic surfaces only approximately match our expectations. These trials are limited by the lack of 3D "truth" at the regional scale, so it is hard

to quantify these mismatches, as we can only compare against our prior concepts of the 3D geology, with all the associated inherent biases. If forced to make a model in these regions, geologists will draw heavily on their expectations, so this form of modelling is not so much a test of their concepts as it is a realisation of them. This opens a pathway to how to deal with conceptual uncertainty is discussed below. It is beyond the scope of this study but very much a topic of interest that may in the future allow these codes to work in a wider range of geological settings.

It is conceivable that we could take these models as starting points for manual refinement of the models, either by adding additional "fictive" data so that the model better matches our pre-conceived notions as geologists, or by exporting the model to a system where manual manipulation of the surfaces is possible. Doing so however defeats the aims of our approach as both approaches introduce modeller-specific biases and void any attempts to use stochastic analyses of alternate parameters, data or feature attribute choices.

### 640 6.2.1 Insufficient data

All geological maps are models, as even in areas of 100% outcrop the map is the sum of hundreds of local observations and interpretations, and in most areas the gaps in outcrop mean that the map can only provide a subset of the potential surface information. It may well be that the surface map does not possess enough information to constrain a 3D model. In many regions, the surface of the Earth is covered by soils or surficial deposits (colluvium, alluvium etc.) that prevent direct

observation of the bedrock geology. In this case there is simply no map to deconstruct. As regional geophysical datasets became more widespread, interpreted maps of the top of bedrock started to be produced (such as the GSWA test case described here), together with estimates of the geometry of the cover-bedrock interface (Ailleres and Betts, 1998). *map2loop* contains example code showing how these may be combined to replace the surface geology as inputs for modelling but were not needed for the Hamersley test case. The integration of geophysics into the workflow is being developed by the Loop consortium (Giraud et

al., 2021), and is beyond the scope of this paper, but could help to define subsurface orientations or even the (automatic?) extraction of geological structures from geophysical data (Wu and Hale, 2015; Vasuki et al., 2017; Wellmann et al., 2017 ; Lindsay et al., 2020; Guo et al., 2021).

Even when surface geology maps are available, interpreted cross-sections are usually added to constrain the 3D geology, however even if they are constrained by geophysical data, by direct interpretation of seismic, or by gravity/magnetic validation

for example, they are still usually less well-constrained than the surface data. Even when seismic data is available, Bond et al. (2015) has shown that this prior experience is a significant source of bias for the interpreted section. Drill hole data are not currently incorporated into the workflow, however the work of Joshi et al., (2021) goes some way to providing that possibility. Geophysically unconstrained cross-sections drawn by geologists necessarily depend on two sources of information, the geology map, in which case in principal a future *map2loop* could provide the equivalent information, or by the geologists'

prior experience, which is harder to codify, and represents a significant future challenge. Many maps indicate a level of

confidence in contacts and fault style via dashed lines, and whilst at present *map2loop* does not make use of this data, it will clearly be an important source of information when incorporating constraints during stochastic simulations. Not all maps follow a chronostratigraphic logic, for example for a map legend of C-B-A (in decreasing age, Fig. 11) a local area of the map may actually show up-sequence orderings of the type C-B-A-B-A-B, and in order for a 3D model to be built they would have to be recoded as C-B1-A1-B2-A2-B3-A3. Of course the repetition of the A-B may be due to deformation (folding of the sequence, or thrust repetition), however it often just represents a level of stratigraphic detail considered unimportant at the scale of the map, or a deliberate avoidance of implying knowledge about the local stratigraphy.

In the early stages of mapping, the locations of contacts can be quite hard to define, so one approach would be to avoid the use of contacts altogether and the SURFE package (Hillier et al., 2014; de Kemp et al., 2017) allows 3D model construction without pre-defined contact locations.

As has been mentioned earlier, in many areas the geology of interest is buried beneath regolith or basins and thus a map-based approach may not be appropriate. Geologists are very good at building models in such data-poor areas, although validation of 3D geological models is often limited to sparse drilling. In this case it is easy to prove that the model is wrong, but much harder to say why.

A second consideration is the actual availability of the data in digital forms. Both within Australia, and internationally, each geological survey has developed its own internal standards for storing and providing outcrop databases, and may do not provide this data at all, except with the map itself. As with the outcrop databases, each country around the world has made their own choices as to the development, or not, of a standard stratigraphy for the country, and the public access to this data. One outcome of increased automation of information extraction from geological maps and other forms of geological data may be the need to establish "minimum data standards" so that the data needed for each type algorithm is made available.

### 6.2.2 Poor quality data

The process of making a map, like any human endeavour, is subject to error, either because of the primary observation, or from the compilation of that information into map form, such as the closed loop fault shown in Fig. 8b. Some analysis of map logic can be made if the information in the input map or stratigraphy is incorrect, such the fault cycles described in Section 3.3.2, although the choice of how to break the cycle is currently arbitrary, a future enhancement may compare the fault relationships with orientation information, for example, to make a better choice. If a 3D model fails to build using the deconstructed data, one may assume there are inconsistencies in the input data. The issue here is the modelling engine will unlikely indicate which data is causing errors, so more robust map validator would be useful that can identify potential issues prior to 3D model input and provide guidance to correction. At present small mismatches between nodes in coincident Polygons and Polylines can be accommodated.

### 6.2.3 Incorrect deconstruction of the data

As discussed in section 4.1, *map2loop* makes a number of simplifications during the deconstruction process. Estimates of fault displacement and unit thickness could be automatically checked for consistency along a contact or fault, which may improve the estimates fed to the modelling schemes.

### 6.2.4 Incomplete 3D modelling algorithms

The last reason that the outputs from *map2loop* do not always produce satisfying 3D geological models is that the modelling systems themselves do not manage all types of geological scenarios well. The modelling engines targeted here are both implicit schemes that work best in regions with a well-defined and gently deformed stratigraphy although *LoopStructural* can also handle poly-deformed terranes. Once overprinting of structures becomes more important, the implicit schemes need more and more information (often provided as interpretations not directly supported by the original data) to reproduce the model

conceived by the geologist. The conceptual model in the geologist's head, what we might call "conceptual priors", is a major control on tuning the implicit model, and codifying these concepts remains a major challenge for the future. To give just one example, the 3D geometry (and even the near-surface dips) of faults are often very poorly understood. In order to produce a 3D model a geologist often brings a preconceived notion: extension related faults offsets with antithetic faults; compression related fault offset with low angle basal fault and associated folds with bedding thickness changes; transpressional and transtensional flower structures, which is then used to complete the model in an under constrained area. All the regional scale tectonic systems (duplex, flower structure etc…) are basically fault networks that evolved with time, with complex slip histories. The LoopStructural library is specifically designed to tackle these sorts of evolutionary systems, however at present the challenge is that we have insufficient data to actually test it in real-world settings. If we could encode these concepts, then it would be easy to ask the automated system to compare model outcomes for the model "as if" it was an extensional listric tectonic environment vs a transtensional system, and a first step to training such an algorithm could analogous to the trained Convolutional Neural Networks of Guo et al. (2020).

One of the keys to improved modelling is to incorporate additional time constraints on the model. All three target modelling engines incorporate some concepts of time, such as stratigraphy-fault age relationships, and LoopStructural can handle superimposed fold and fault interference geometries if sufficient data is available (Grose et al., this volume). Finally, the choice of which data to put into the 3D model is by definition outside of the 'knowledge' of *map2loop*, as it can only process datasets it has been made aware of, however a broader data discovery algorithm that searched for all available data and then decided on the basis of, for example, data density, relevance to question, volume of interest (Aitken et al., 2018) could be a way to avoid this currently biased process.

## 6.3 Future work

The enormous advantage of automating many of the somewhat arbitrary choices and calculations described in this paper is that alternatives can also be coded, and the sensitivity of the resulting 3D models to these choices could be analysed. A beta version of a stochastic model ensemble generator containing elements of the work presented in Wellmann and Regenauer-Lieb (2012); Lindsay et al. (2012); Pakyuz-Charrier et al. (2018a&b) and de la Varga and Wellmann (2016) is under development (https://github.com/Loop3D/ensemble_generator). Since the process is automatic, the time taken to calculate 1000 models on a distributed computing system is the same as calculating one model, so very large model suites can be explored for very little additional time cost. This can build on existing capabilities: *GemPy* has its own advanced framework for analysing uncertainty (de la Varga et al., 2019). Work is currently underway to wrap the entire data extraction, 3D geological modelling and geophysical forward and inverse modelling workflow in Bayesian analysis framework, so that the distinct and cumulative effects of all modelling, uncertainty quantification and joint geological-geophysical inversion decisions (e.g. Giraud et al., 2020) can be analysed in a homogeneous fashion. Other current studies, as mentioned earlier, include building libraries for automating information extraction from drill hole data (Joshi et al., 2021) and the inclusion of sill-like intrusive contacts (Alvarado-Neves et al., 2020.

In the immediate future *map2loop* and related codes need to manage a wider range of input datasets including drill holes and cross sections, and this work is underway. There is also a need to extract the maximum amount and range of information from other igneous intrusions that do not follow simply stratigraphic or geometric rules. Perhaps the biggest challenge is the incorporation of conceptual constraints during the deconstruction workflow, as discussed in the previous section (Jessell, 2021).

## 7. Conclusions

The automation of map deconstruction by *map2loop* provides significant advantages on manual 3D modelling workflows, since it:

- Significantly reduces the time to first prototype models, from hours to minutes for the example shown.
- Allows reproducible modelling from raw data since the data extraction, decimation and calculation parameters are defined up front by the user.
- Clearly separates the primary observations, interpretations, derived data and conceptual priors during the data reduction steps and
- Provides a homogenous pathway to Sensitivity Analysis, Uncertainty Quantification, Value of Information studies.

## 8. Author Contribution

Mark Jessell was responsible for implementation of the computer code and supporting algorithms for the *map2loop* and wrote the initial draft of the manuscript. Vitaliy Ogarko was responsible for implementation of the computer code and supporting algorithms for the *map2model* code. Yohan de Rose was responsible for refactoring the code and suggesting and implementing improvements to the efficiency of the code. Mark Lindsay tested the code on different datasets and contributed to code design and manuscript pre-submission revision. Ranee Joshi implemented parts of the computer code and supporting algorithms for the *map2loop* and contributed to manuscript pre-submission revision. Agnieszka Piechocka tested the code on different datasets and contributed to code design and contributed to manuscript pre-submission revision. Lachlan Grose implemented parts of the computer code and supporting algorithms for the *map2loop* and contributed to manuscript pre-submission revision. Miguel de la Varga implemented parts of the computer code and supporting algorithms for the *map2loop* and contributed to manuscript pre-submission revision. Laurent Ailleres contributed to code design and manuscript pre-submission revision. Guillaume Pirot tested the code on different datasets and contributed to code design and manuscript pre-submission revision.

## 9. Acknowledgements

We acknowledge the support from the ARC-funded Loop: Enabling Stochastic 3D Geological Modelling consortia (LP170100985) and DECRA (DE190100431). The work has been supported by the Mineral Exploration Cooperative Research Centre whose activities are funded by the Australian Government's Cooperative Research Centre Programme. This is MinEx CRC Document 2021/9. Source data provided by GSWA and Geoscience Australia. We would like to thank Rob Harrap and Stuart Clark for their thorough reviews that significantly improved the manuscript.

10. Appendices

**Appendix1. Minimum required inputs for *map2loop***

**Minimum *map2loop* inputs:**

1. EPSG coordinate reference system for input data (e.g. metre-based projection like UTM)

2. Max/min coordinates of area of interest

3. Geology Polygons:

   -a. All Polygons are watertight (node location mismatches must be within a smaller definable error)

   -b. Polygons have as attributes:

         -i. Object ID

-ii. Stratigraphic code

         -iii. Stratigraphic group

         -iv. One of more fields that describe if sill, if igneous, if volcanic

         -v. Min_age field

         -vi. Max_age field (can be same as Min_age field, and can be simple numerical ordering (bigger number is

older))

4. Fault/Fold Axial Trace Polylines:

   -a. Faults terminate on other faults but do not cross

   -b. Faults/Folds have as attributes:

-i. Object ID

         -ii. Field that determines if Polyline is fault or fold axial trace

         -iii. Field that determine type of fold axial trace (e.g. syncline or anticline)

         -iv. Faults can have dip/dip direction info

5. Bedding orientations:

-a. Assumes dip/dip direction or dip/strike data

   -b. Orientations have as attributes:

         -i. Dip

         -ii. Dip Direction or strike

## Appendix 2. Pseudocode for key calculations

**save_basal_contacts**

explode geology polgyons so interior holes become distinct Polygons

       for each Polygon:

         build list of Polygons and their 23odelling23

       load sorted stratigraphy from csv file

       for each Polygon in list:

if not intrusive:

         if Polygon Code found in sorted stratigraphy:

          for each Polygon in list:

           if two Polygons are not the same:

            if two Polygons are neighbours:

if second Polygon is not a sill:

              add neigbour to list

          if first Polygon has neighbours:

           for each neighbour:

           if neighbour Polygon Code found in sorted stratigraphy:

if neighbour older than first Polygon:

             calculate intersection of two Polygons:

              if intersection is a multilinestring:

               for all line segments in linestring:

                save out segment with x,y,z Code

build dictionary of basal contacts and dictionary of decimated basal contacts

       return dictionary of basal contacts and dictionary of decimated basal contacts

**save_basal_no_faults**


       load fault linestrings as GeoDataBase

       create Polygonal buffer 23odell all faults

       clip basal contacts to Polygonal buffer

       make copy of clipped contacts

for each clipped basal contact Polyline:

        if Polyline is GEOMETRYCOLLECTION:

         remove from copy of clipped basal contacts

        else:

         add to dictionary


       build GeoDataFrame from remaining clipped basal contacts and save out as shapefile

**save_fold_axial_traces_orientations**

load geology Polygons as GeoDataFrame

       load interpolated contacts as array

       load Polylines as GeoDataFrame

       for each Polyline:

        for each line segment in Polyline:

if fold axial trace:

          if passes decimate test:

           calculate azimuth of line segment

           calculate points either side of line segment

           find closest interpolated contact

if interpolated contact is sub-parallel to fold axial trace:

             save orientation data either side of segment and related x,y,z,Code to csv file

       **interpolate_contacts**

create grid of positions for interpolation, or use predefined list of points

         for each linestring from basal contacts:

          if passes decimation test:

           for each line segment in linestring:

             calculate direction cosines of line segment and save to file as csv with x,y,z,etc


         interpolate direction cosines of contact segments

         save interpolated contacts to csv files as direction cosines and azimuth info with x,y,z,etc

**interpolate_orientations**

         subset points to those wanted

         create grid of positions for interpolation, or use predefined list of points

         for each point from orientations:

calculate direction cosines of orientations

         interpolate direction cosines of orientations

         save interpolated orientations to csv files as direction cosines and dip,azimuth info with x,y,z,etc


       **join_contacts_and_orientations**

         for each orientation in grid:

          rescale contact direction cosines with z cosine of orientations

save out rescaled x,y direction cosines from contacts with z direction cosine from orientations and positional x,y,z,Code

       **calc_thickness**

         load basal contacts as vectors from csv file

load interpolated bedding orientations from csv file

         load basal contacts as geopandas GeoDataFrame of Polylines

         load sorted stratigraphy from csv file

         calculate distance matrix of all orientations to all contacts

for each contact line segment:

          if orientations within buffer range to contact:

           calculate average of all orientation direction cosines within range

           calculate line normal to contact and intersecting its mid-point

           for all basal contact Polylines:

if Polyline Group is one stratigraphically one unit higher:

             if contact normal line intersects Polyline:

               if distance between intersection and contact mid-point less than 2 x buffer:

                 store info

           from list of possible intersections, select one closest to contact mid-point

if closest is less than maximumum allowed thickness:

            save thickness and location to csv file

**11. Code availability**

*The map2loop & map2model* codes are available with a MIT Licence. The link below provides access to the source code and
brief user guide.

http://doi.org/10.5281/zenodo.4288476

Example Jupyter notebooks (Examples 1,2,3) that work with online or user-supplied datasets are available here

Zenodo link to notebooks.

**12. Data availability**

http://doi.org/10.5281/zenodo.4288476

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

**Table 1. Parameters that may be modified from their defaults prior to the automated workflow starting.**

| Parameter name | Meaning | Default value | Data type |
|---|---|---|---|
| aus | Indicates if area is in Australia for using ASUD | TRUE | bool |
| close_dip | Dip to assign to limbs of folds, -999 means use interpolated dip as local dip estimator, otherwise apply fixed dip assuming normal younging. | -999 | int |
| contact_decimate | Save every nth contact data point | 5 | int |
| contact_dip | Contact dip information, -999 means use interpolated dip as local dip estimator, otherwise apply fixed dip assuming normal younging. | -999 | int |
| contact_orientation_decimate | Save every nth contact orientation point | 5 | int |
| deposits | Mineral deposit commodities for focused topology extraction. Not discussed int this paper. | 'Fe,Cu,Au,NONE' | str |
| dist_buffer | Buffer for processing plutons to ensure faults that stop at plutons are correctly analysed by *map2model* | 10 | int |
| dtb | Path to depth to basement grid | '' | str |
| fat_step | How much to step out normal to the fold axial trace for limb orientation to be added | 750 | int |
| fault_decimate | Save every nth fault data point | 5 | int |
| fault_dip | Default fault dip , -999 means add randomly assigned value between +/- 60 degrees | 90 | int |
| fold_decimate | Save every nth fold axial trace data point | 5 | int |
| interpolation_scheme | What interpolation method to use of *scipy_rbf* radial basis or *scipy_idw* inverse distance weighting | 'scipy_rbf' | str |
| interpolation_spacing | Interpolation grid spacing in meters | 500 | int |
| intrusion_mode | 0 to only exclude sills, 1 to exclude all intrusions from basal contacts | 0 | int |
| max_thickness_allowed | When estimating local formation thickness, make upper limit to valid thicknesses to avoid unlikely thickness values | 10000 | int |
| min_fault_length | Min fault length (tip to tip straight line distance) to be used | 5000 | int |
| misorientation | Maximum misorientation of pole to great circle of bedding between stratigraphic groups to be considered part of same supergroup | 30 | int |
| null_scheme | Value of null values (i.e. surface outcrop) in the depth to basement grid | 'null' | str |
| orientation_decimate | Save every nth orientation data point | 0 | int |
| pluton_dip | Default pluton contact dip | 45 | int |
| pluton_form | Possible forms from 'domes', 'saucers' 'pendants', 'batholiths' | 'domes' | str |
| thickness_buffer | How far away to look for next highest unit when calculating formation thickness | 5000 | int |
| use_fat | Use fold axial trace info to add near-axis bedding info | TRUE | bool |
| use_interpolations | Use all interpolated dips for modelling | TRUE | bool |

**Table 2. Geometric features imported and saved by *map2loop* and *map2model*. The geometric objects refer to specific *Geopandas* data objects.**

| Geometric Object | Input Geological Feature | Augmented Output Geological Feature |
|---|---|---|
| Point | Bedding | Bedding, Contacts, Faults, Fold Axial Traces |
| Polyline | Faults, Fold Axial Traces | None |
| MultiPolyline | Faults, Fold Axial Traces | None |
| Polygon | Stratigraphic domains | None |
| MultiPolygon | Stratigraphic domains | None |
| Raster | DTM | DTM |


**Table 3 Comparison between model engine inputs**

| Modelling Engine | Digital Terrain Model | Stratigraphy | Orientation data | Stratigraphic units | Faults | Fold axial traces |
|---|---|---|---|---|---|---|
| *LoopStructural* | Used | 2-level | Bedding, Cleavages | Position, thickness of units | Position, age relationships wrt units and each other, displacement, ellipsoid for limited extent faults | Not used directly |
| *GemPy* | Used | 2-level | Bedding | Position | Position, age relationships wrt units and each other, displacement, ellipsoid for limited extent faults | Not used directly |

**Table 4. Augmented outputs provided by *map2loop/map2model*. Many other outputs are not described here are not currently used by the target modelling engines, and some simply provide debugging information.**

| Data type | Content | File path |
|---|---|---|
| Position | georeferenced dtm | dtm/dtm_rp.tif |
| Position | Contact info with z and formation | output/contacts_clean.csv |
| Position | Contact info with tangent info | tmp/raw_contacts.csv |
| Position | Fault trace with z | output/faults.csv |
| Position | Local formation thickness estimates | output/formation_thicknesses_norm.csv |
| Position | Fault dimensions | output/fault_dimensions.csv |
| Position | Fault displacements | output/fault_displacement3.csv |
| Gradient | Fault orientation with z | output/fault_orientations.csv |
| Gradient | Bed dip dd data with z and formation | output/orientations_clean.csv |
| Topology | Summary stratigraphy relationships | tmp/all_sorts_clean.csv |
| Topology | Fault-fault relationship table | output/fault-fault-relationships.csv |
| Topology | Fault-fault relationship graph | output/fault_network.gml |
| Topology | Fault-group relationship table | output/group-fault-relationships.csv |
| Topology | Sets of structurally coherent groups | tmp/super_groups.csv |
| Topology | Fault-relationship graph | tmp/fault_network.gml |
| Program Control | Bounding box of model | tmp/bbox.csv |


**Table 5. Time taken to manually reproduce the step taken by the automated process. The addition of z values can be managed by the 3D modelling packages, so the time to perform this task manually is not included, except where it is needed in the calculation (calculation of true formation thickness).**

| Task | Timing of Manual Task (minutes) | Breakdown of activities |
|---|---|---|
| DTM | 13 | convert ROI coordinates to Lat/Long; download SRTM tile; reproject; save as geotif |
| basal contacts | 44 | Re-digitise basal contacts; save as csv |
| bedding orientations | 15 | add formation info; save as csv |
| fault offsets | 63 | locate measurable offsets; estimate local bedding dips; calculate true offset for vertical displacement; save as csv |
| formation thicknesses | 46 | chose bed thicknesses to calculate; estimate local bedding dips; calculate true thickness; save as csv |
| faults | 13 | simplify fault polylines; save as csv |
| fault-fault | 14 | identify fault-fault intersections; build fault topology graph; save as csv matrix |
| fault-strat | 18 | identify fault-stratigraphy intersections; build fault topology graph; save as csv matrix |
| build stratigraphic table | 16 | get stratigraphy from map legend as simplify to match roi of interest; save as csv |
| Contact info with tangent info | 18 | calculate local normal; add xy; save to csv |
| Fault dimensions | 7 | calculate fault length; save to csv |
| Fault orientation with z | 11 | calculate fault orientation; add dip; add xy; save to csv |
| Sets of structurally coherent groups | 6 | Make supergroup table |
| **Total** | **284** | |


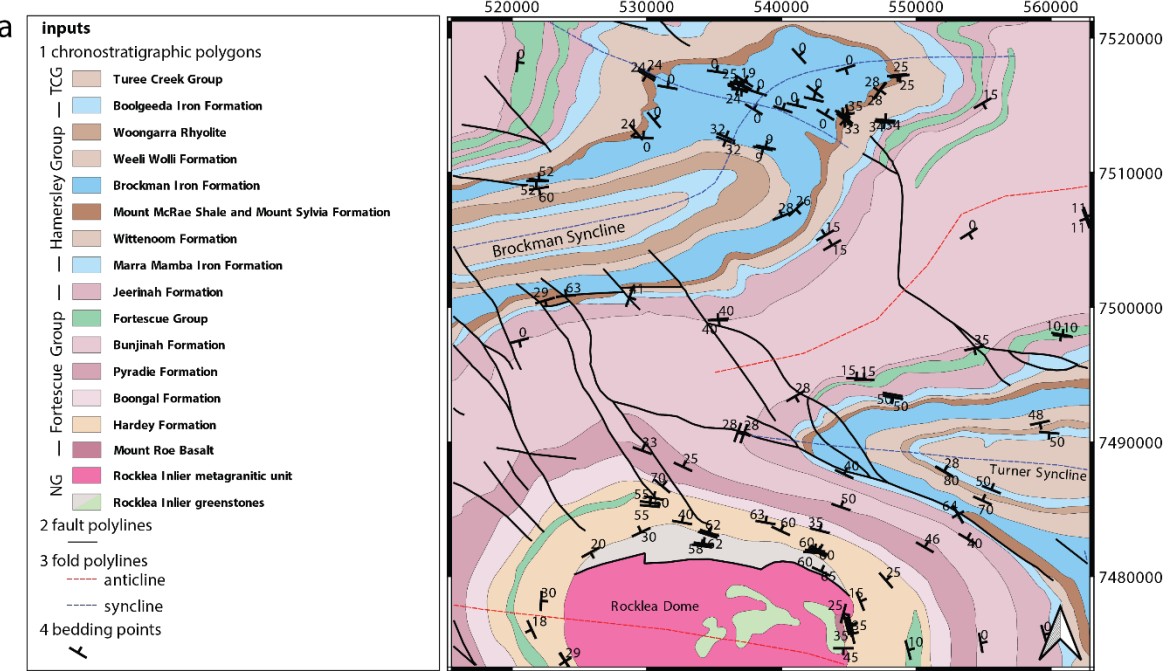

b   **5 Stratigraphic (overlies ->) Relationships**

Turee Creek Group **->** Boolgeeda Iron Formation
Wittenoom Formation **->** Marra Mamba Iron Formation
Marra Mamba Iron Formation **->** Jeerinah Formation
Jeerinah Formation **->** Bunjinah Formation
Bunjinah Formation **->** Pyradie Formation
Pyradie Formation **->** Boongal Formation
Weeli Wolli Formation -> Brockman Iron Formation

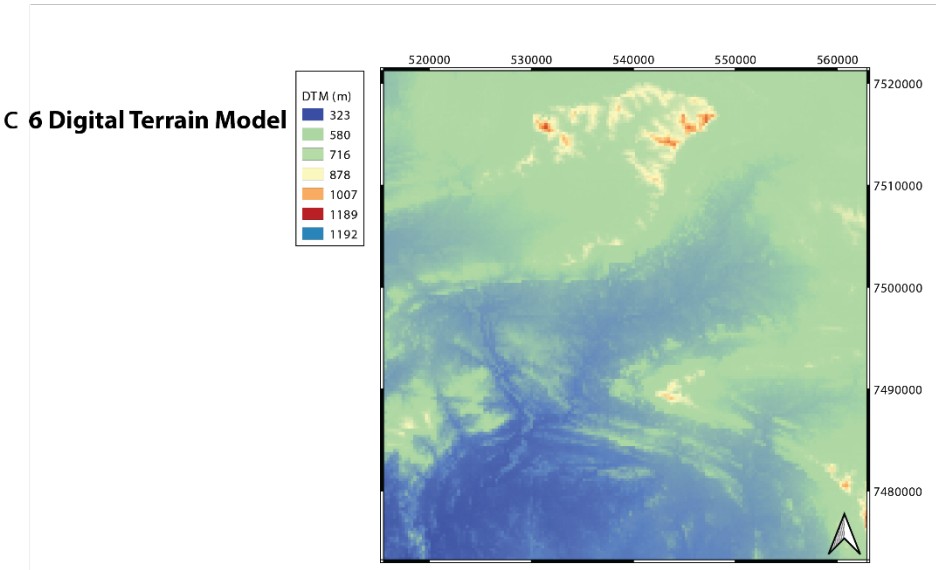

**Figure 1. The six types of inputs to *map2loop*. a) 1:500,000 Interpreted bedrock geology of the Rocklea Dome region of Western Australia showing the different datasets used to create the 3D model. TCG, Turee Creek Group. NG, no group defined by map, so each unit is its own group. The region shown is approximately defined by the max/min lat/long coordinates [ 117.15, -22.84, 117.60, -22.41 ]. b) First seven entries of the binary stratigraphic relationships derived from the Australian Stratigraphic Units Database that relate to the test area (ASUD, Geoscience Australia and Australian Stratigraphy Commission. (2017). Australian Stratigraphic Units Database). c) The SRTM digital terrain model is sourced directly from Geoscience Australia at: http://services.ga.gov.au/gis/services/DEM_SRTM_1Second_over_Bathymetry_Topography/MapServer/WCSServer**

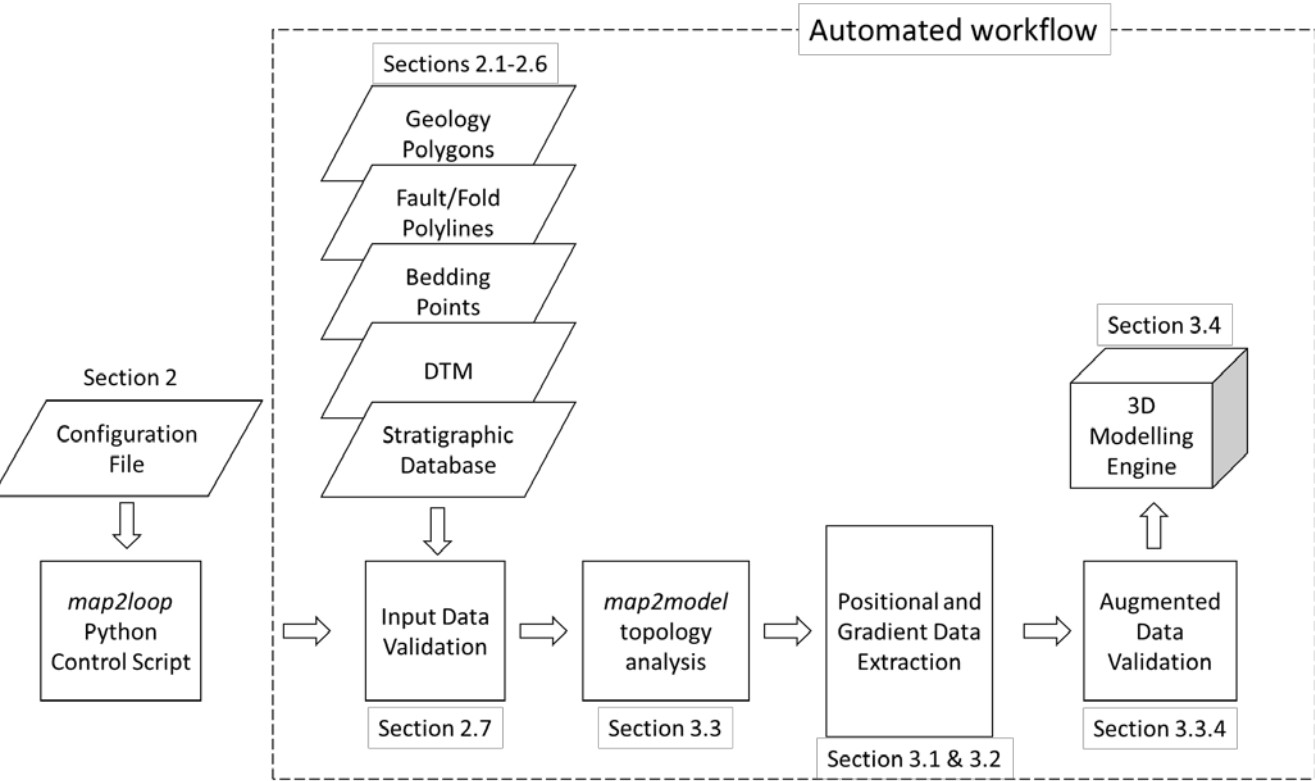

Figure 2. Automated workflow. Once the Configuration File has been created, and the workflow parameters (Table 1) have been defined in the *map2loop* Control Script, all steps within the dashed rectangle are fully automated, with no manual intervention, These automated steps are described in the associated sections, from accessing the data through to and including the construction of the 3D geological model with *LoopStructural* or *GemPy*. Note that this is a schematic workflow, as individual steps need to be performed out of sequence for computational efficiency. A more detailed workflow is shown in Figure 5.

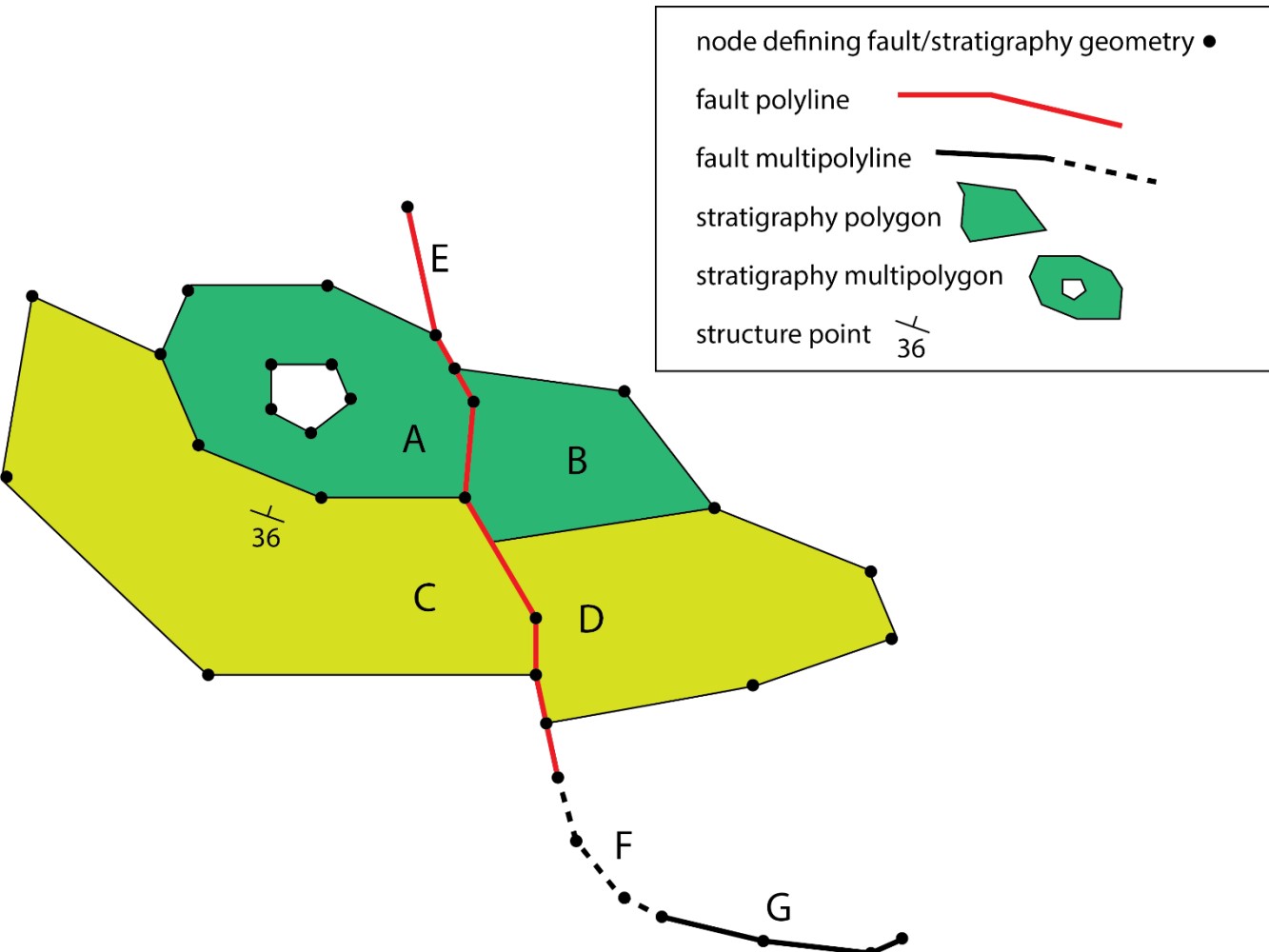

**Figure 3. Geometric elements used in geological maps. B, C & D are stratigraphic Polygons, defined by a sequence of the x,y locations of nodes. A is a MultiPolygon as it contains a hole, although MultiPolygons can also describe two unconnected Polygons. E is a fault Polyline. F & G are fault MultiPolylines that describe segments of the same fault (as does fault E in this case). The structure observation (bedding measurement) is of type Point. All geometric elements may possess multiple attributes and are converted to 3D equivalents by add the information from the raster DTM.**


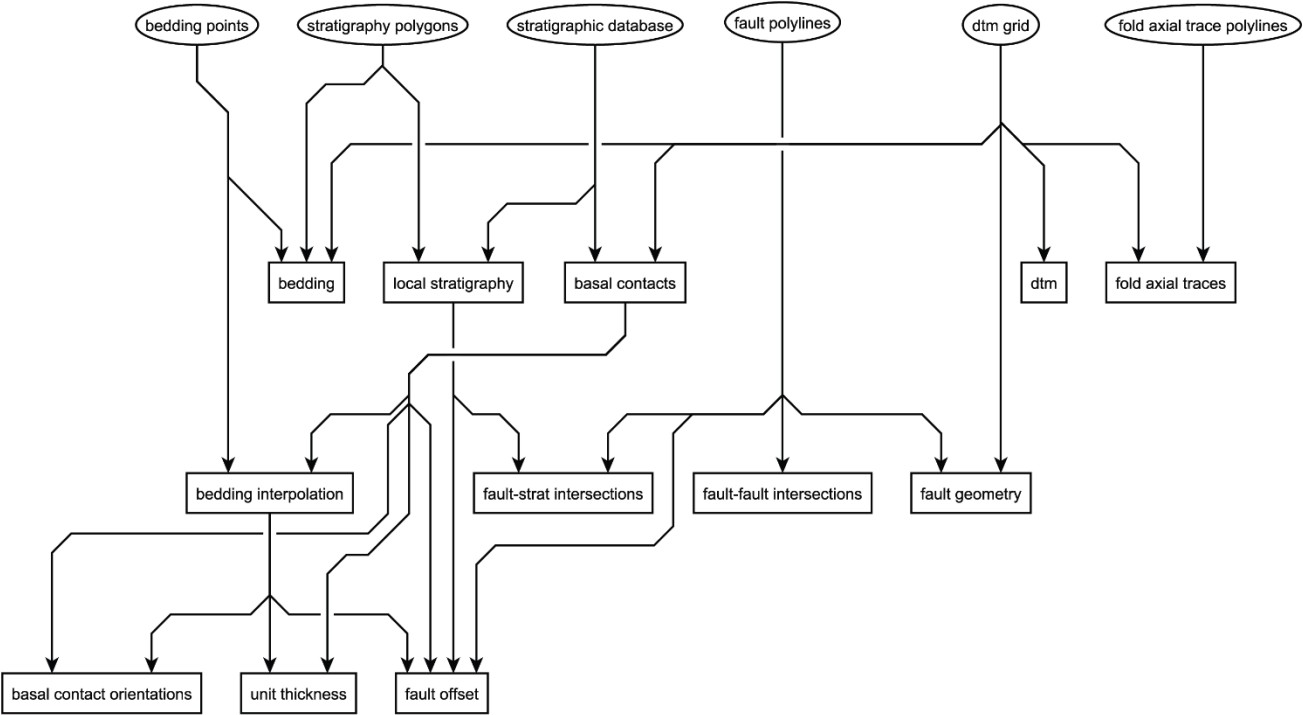

**Figure 4. Data flow from inputs (ellipses) provided by GIS map layers, web servers, and stratigraphic databases. Augmented data (rectangles) are calculated by combining the inputs directly or incrementally during the *map2loop* workflow. The *map2model* code handles the topological analysis: fault-fault intersections, fault-stratigraphy intersections, and local stratigraphic analysis, all other calculations are managed by *map2loop*.**

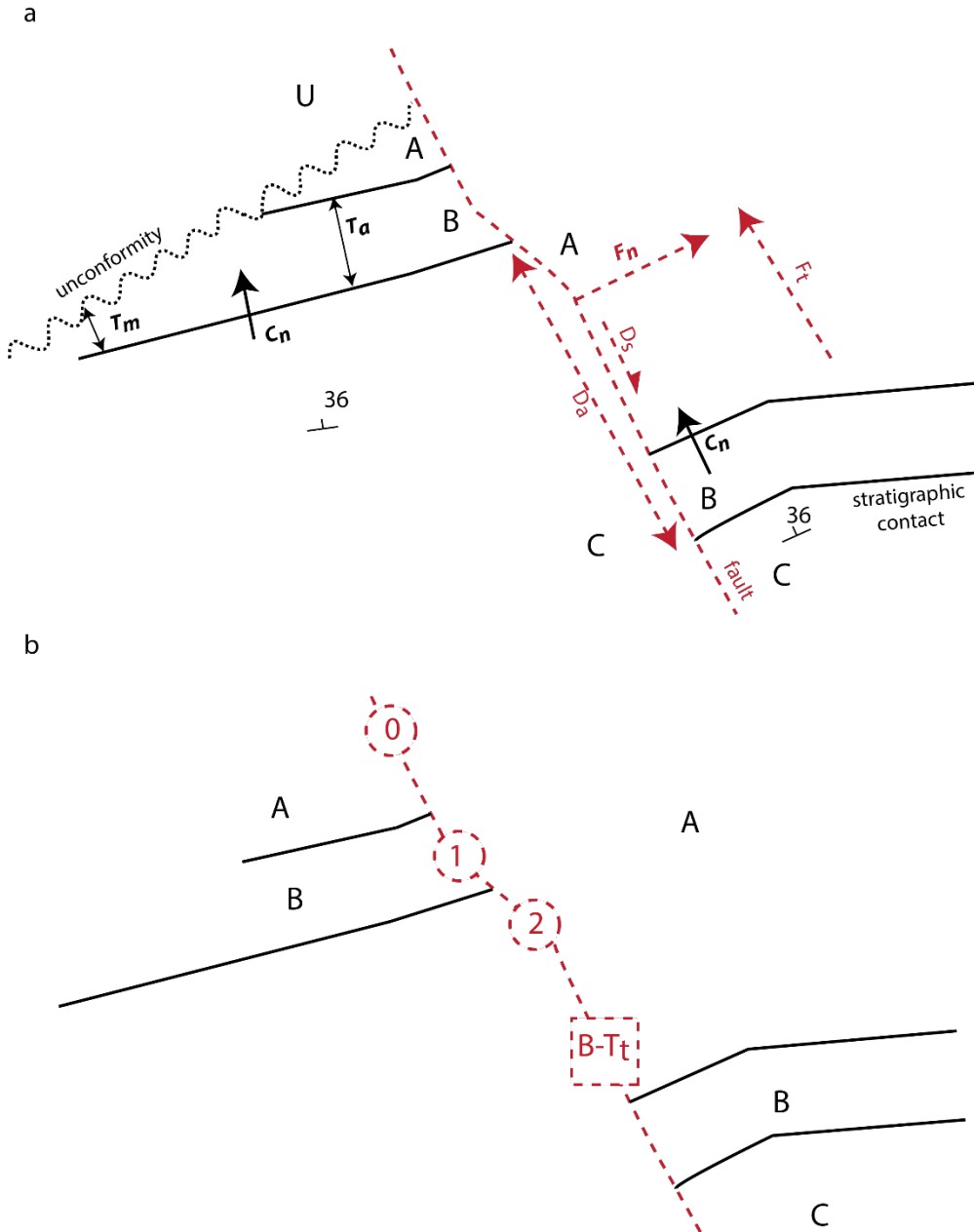


**Figure 5. Positional calculations. a) Apparent unit thicknesses are calculated by calculating the normal distance from a contact ($T_a$) and are then transformed to 'true' thicknesses as by considering the local dip of the bedding. Apparent displacement is calculated by matching equivalent contacts across the fault, in this example the B-C contact ($D_a$). This is then transformed to 'true' displacement by assuming a down-dip slip vector. Finally, the downthrown direction is calculated by examining the cross product**
**of the fault trace ($F_t$) and the dip direction of the strata multiplied by the displacement. See text for details. b) If the direct calculation of fault displacement is not possible, because equivalent contacts across the fault cannot be established, then a minimum displacement can be estimated by the stratigraphic offset in terms of unit thicknesses. In the example here, the dashed red square indicates that the fault locally separates units A and C, so the minimum displacement is the thickness of unit B, which we were able to calculate above. If the unit thickness is not calculable for some reason, the stratigraphic offset between units A-A, A-B and A-C**
**indicate a stratigraphic offset of 0, 1 and 2 stratigraphic units (red dashed circles).**

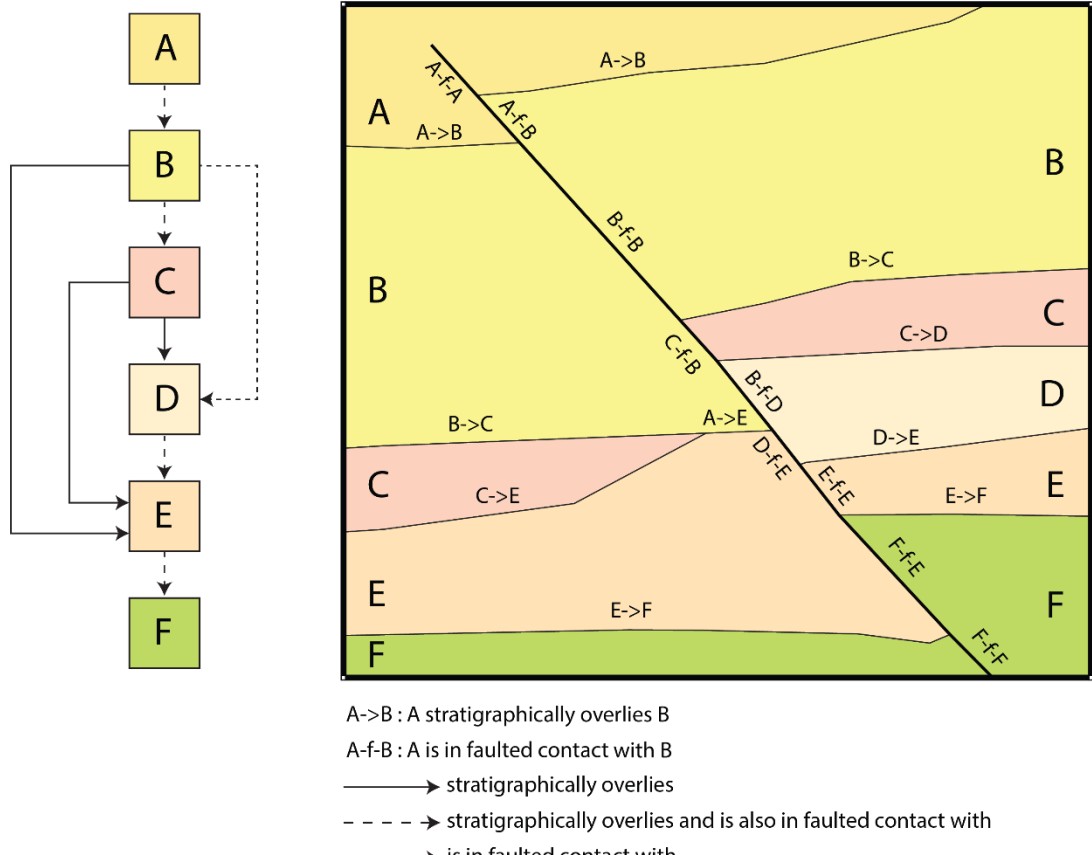

A->B : A stratigraphically overlies B

A-f-B : A is in faulted contact with B

⟶ stratigraphically overlies

– – – ⟶ stratigraphically overlies and is also in faulted contact with

-------- ⟶ is in faulted contact with

**Figure 6. Example topological relations extracted from the map by the *map2model* library. In this map we have 6 units A-F which locally are in contact with each other either by normal stratigraphic relationships (A->B signifying that A is younger), or separated by a fault (A-f-B with no relative age significance). Once these individual binary relationships are aggregated by *map2model* into a single graph, specific pairs of units may be stratigraphic only (solid line), a combination of stratigraphic and fault relationships (long dashed line) or fault-only (short dashed line). Intrusive relationships (not shown here) will also be extracted from the map where present. These relationships form the basis of our understanding of the local stratigraphic graph.**


a. contacts

b. faults

c. folds

d. unit thickness

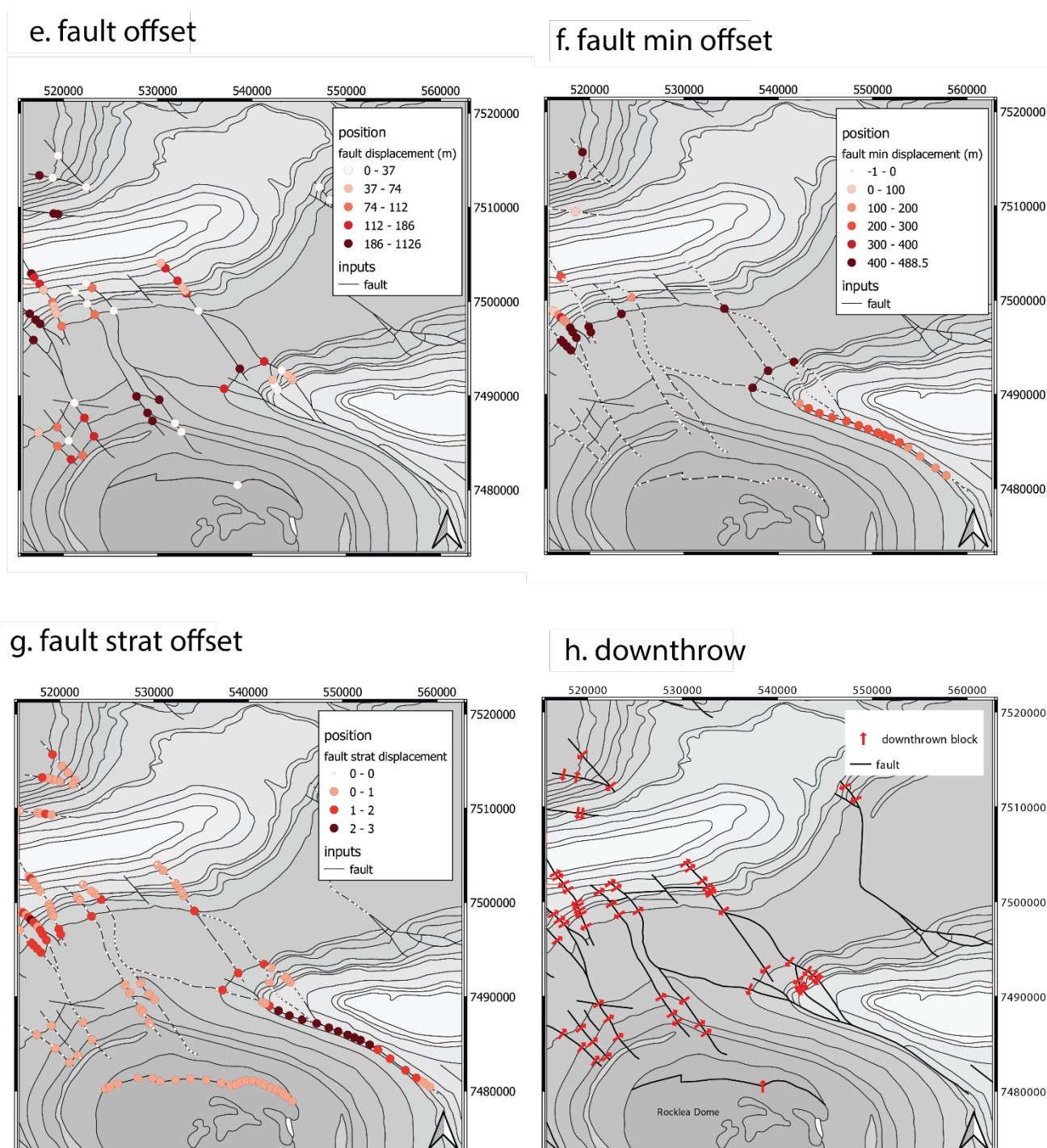

**Figure 7. Positional information derived from map: a) Basal contacts of stratigraphic units, colours as Figure 1. b) Fault traces, colours randomly assigned to each fault, only faults longer than a defined length, in this case 5km, are processed. c) Fold axial traces, d) Local unit thicknesses. e) Fault offset, assuming down-dip displacement. f) Fault offset derived from minimum stratigraphic offset g) Stratigraphic fault offset and h) Fault downthrown block direction.**


## a. fold orientations

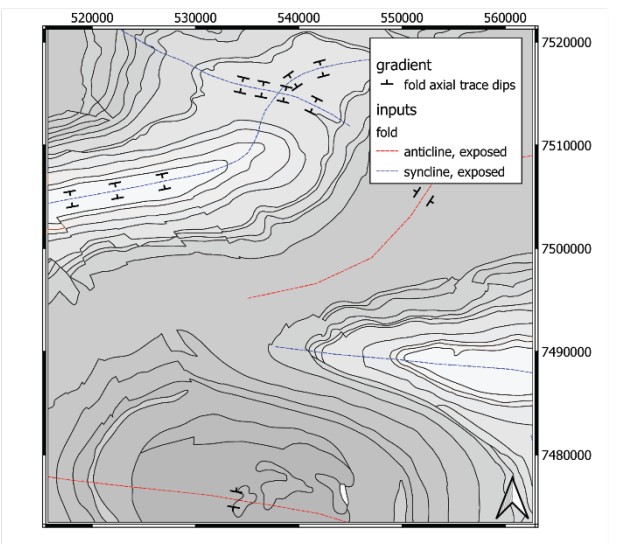

## b. fault orientations

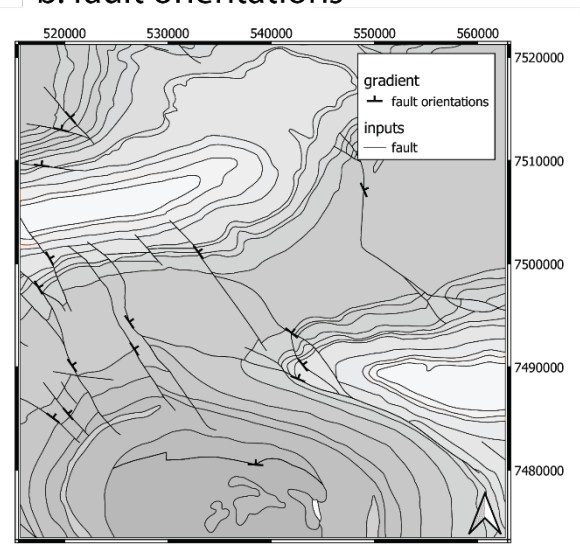

## c. orientation field

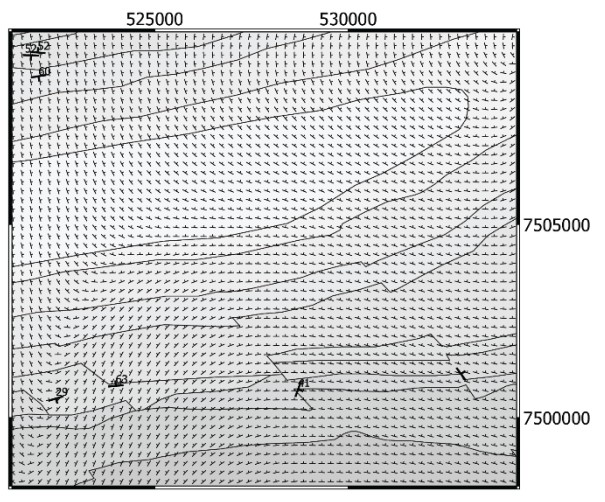

## d. contact field

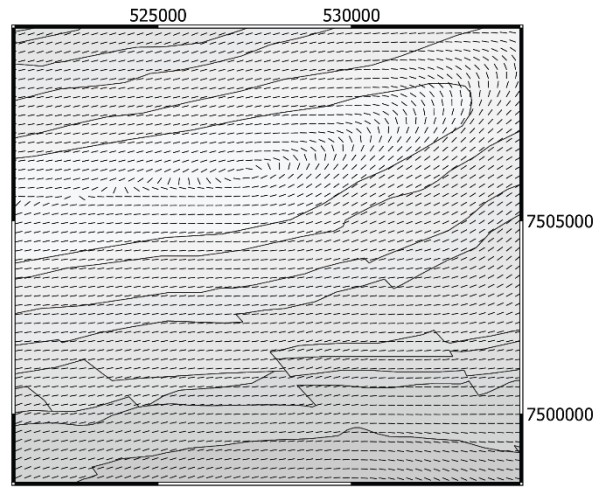

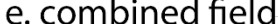

## e. combined field

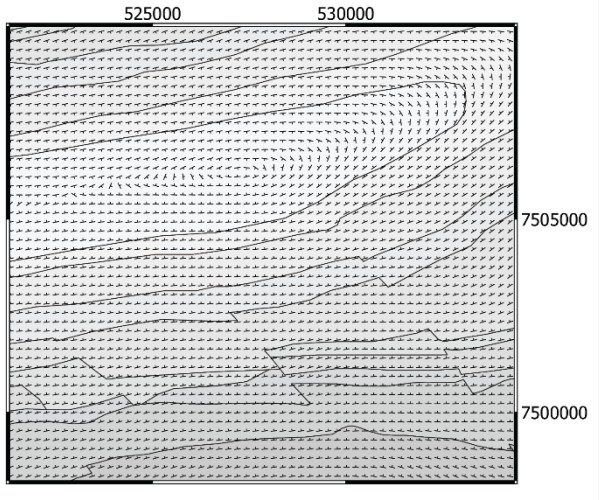

**Figure 8. Gradient information derived from map, zoomed into to Brockman Syncline: a) Bedding orientations near fold axial traces.**
**b) Fault orientations. c) Interpolated orientation data, calculated as interpolated $l_c, m_c$ , inset of part of NW area of map. d)**
**Interpolated contact tangents, calculated as interpolated $l_o, m_o, n_o$ direction cosines, inset of part of NW area. e) Combined**
**information from interpolated dips and interpolated contacts, inset of part of NW area.**


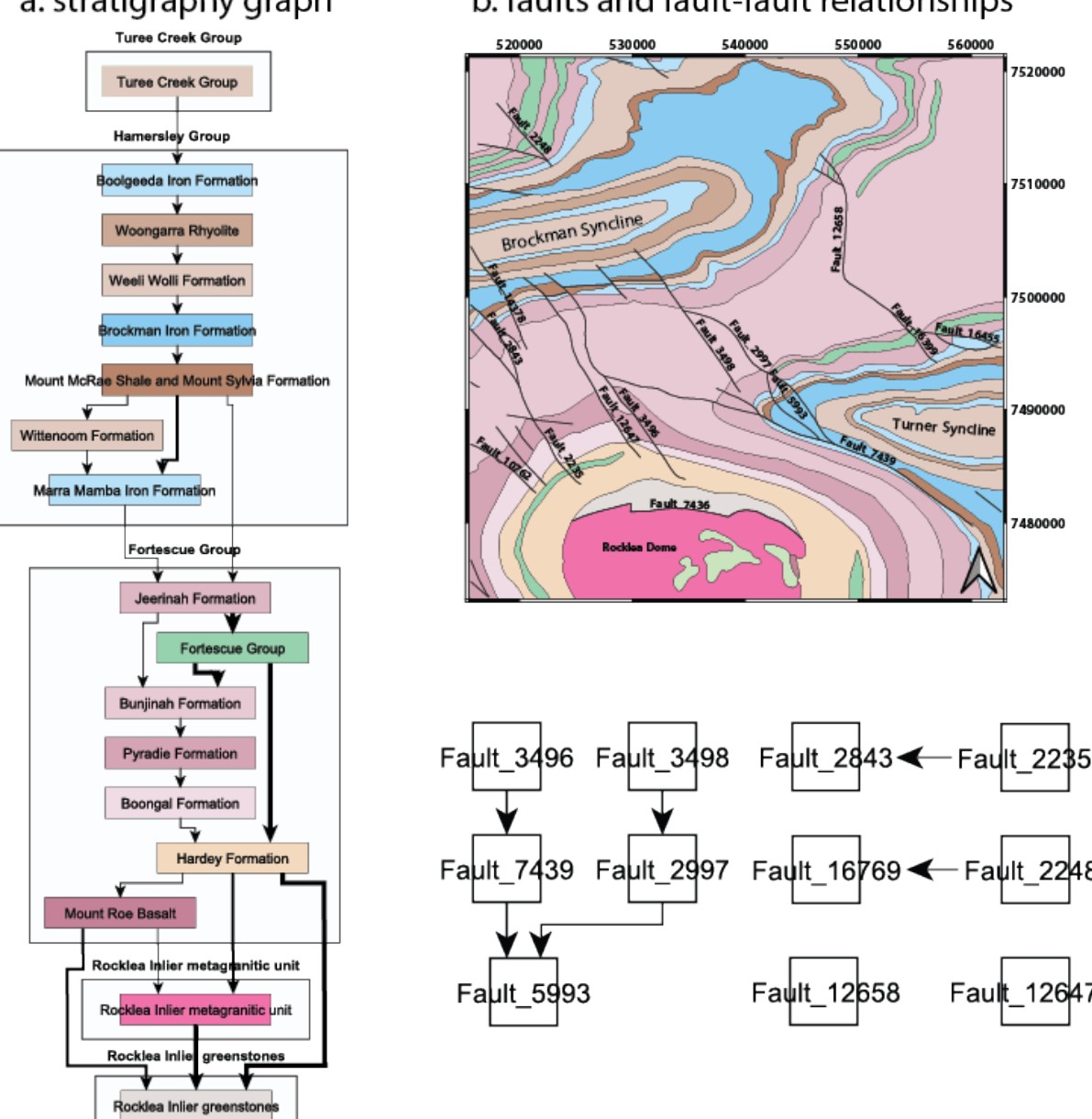

**Figure 9. Topological information derived from map: a) Stratigraphic ages relationships extracted from map and ASUD. Arrows point to older unit. Thickness of arrows is proportional to contact length. b) Map with fault labels for faults longer than 5km, and below the map the resulting fault-intersection relationships. Arrows point to older fault. c) Subset of fault-unit truncation relationships, green cells show stratigraphic units cut by faults, yellow cells show that the unit is not cut by a given fault.**

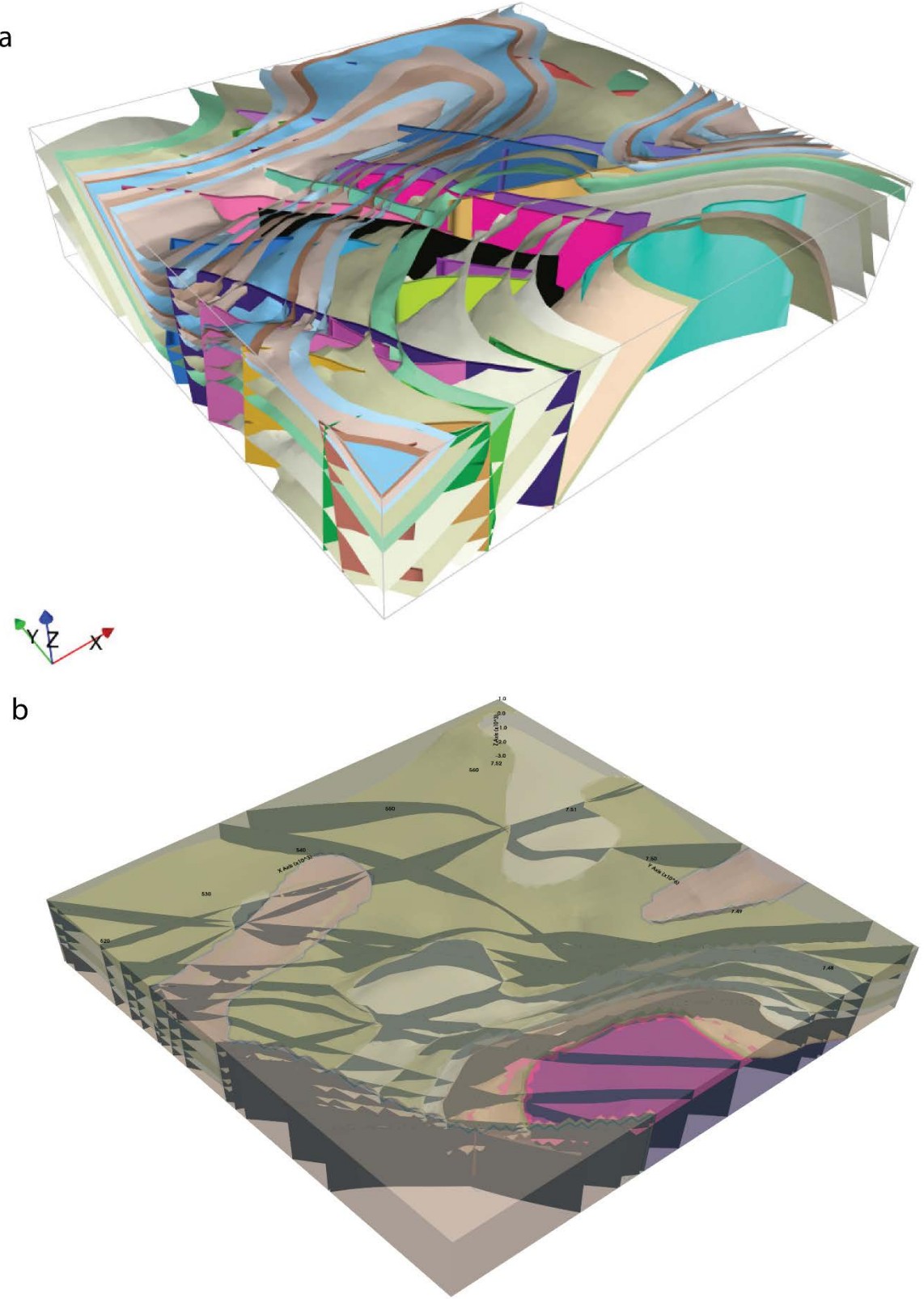


**Figure 10. 3D Models built by a)** *LoopStructural* **and b)** *GemPy* **using the augmented data provided by** *map2loop*. **Note that different packages use different subsets of the available data.** *GemPy* **calculates limited-extent faults but currently displays them as extending across the model area.**

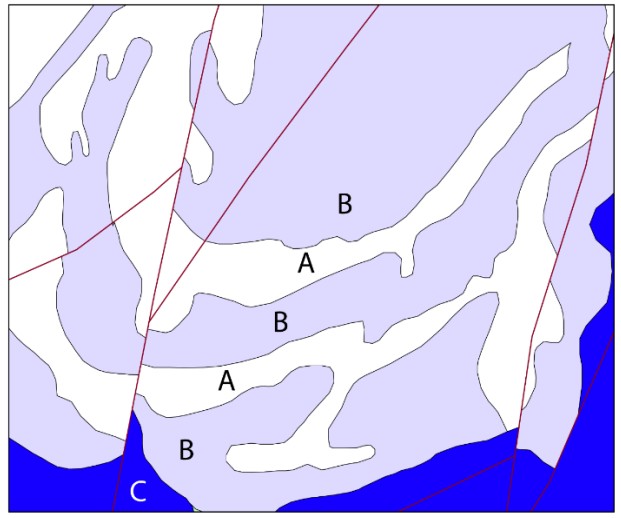

**Figure 11. Example of lithological map descriptions that need recoding in order to work in a chronostratigraphic modelling workflow. Assuming that the repetition of units is not structurally controlled, the lithostratigraphic sequence C-B-A-B-A-B-A needs to be recoded as C-B1-A1-B2-A2-B3-A3.**