# Peer review of "Automated geological map deconstruction for 3D model construction using *map2loop 1.0* and *map2model 1.0"

_Geoscientific Model Development, 2020_

## Referee Comment (RC1)

**Reviewer comments – Automated Geological Map Deconstruction for 3d Model Construction**

R.M. Harrap, Queen's University, Kingston, Ontario.
harrap@queensu.ca

**General Comments**

This is an important contribution to an area of growing importance and relevance. The paper thoroughly describes a tool which is significant both in terms of general approach and highlighting areas of map to model automation that are or are not challenging, and also exposing the areas in which geological mapping and the thinking process that geologists go through show significant ambiguity that plagues later use of that data. While the paper is quite focused on the tool, the deconstruction process and the discussion of areas in which the tool can (or perhaps cannot) be improved are of general interest.

My only general objection to the form of the paper as it now stands is that there is no description of how the tool is actually run – is it an iterative process with a human intervening, or a batch process where a model run that fails or succeeds can be rerun, or… exactly what? Many tools we use in geology, and perhaps those that are the most influential, are used as what-if thinking tools in a way that becomes second nature to an experienced user. I'd like to see some discussion of how the tool fits into the workflow of a geologist/modeler 'as a tool.'

I highly recommend the paper be published with minor corrections. Thank you for the opportunity to be involved in the process.

**Scientific Specific Comments and Requests for Clarification**

Some of these are quite minor points, some may reflect me misunderstanding the paper (which says something) and some are fairly technical. The format is line number or range followed by the point, however large or trivial.

30: Why say deformed? Doesn't it work on flat-lying strata?

35: The point of this work (I'm the author) was consistency checking both *during* and *after* map creation and especially when large, complex compilation maps were being created and to focus on areas where a legend contradicts map relationships. This was implemented at one point in an unpublished MSc thesis and it worked. The point here is that workers such as Brodaric and Harrap were always interested in tools used *during* mapping as well as *during compilation* and not simply in analyzing others maps. Not sure this requires any change in the paper, but I thought it interesting to point out given the long and slow history of developments of tools in this area!

43: I'm not sure what irreproducible means here.

46: Is that transformation unique? Or is it an interpretation especially in cases where legends are being adapted? Brodaric, and Coleman-Sadd (iirc) did some work on reconciling maps where sub-maps had different authors and different emphasis during mapping.

51: As well as for use in other studies?

54: … lack stratigraphic information. This reads oddly since drill holes in non-hardrock terrain would never lack stratigraphic information? Perhaps I'm misunderstanding here.

55: The 3 types specified here – might be expanded with one sentence of more detail on each? And why (just) these three? As you'll see below I have comments about others though you address that very well in your Discussion in the end.

58: Combine direct with conceptual?
   a) What is knowledge from nearby areas?
   b) Tectonic history (okay, you cover this)
   c) What about base scientific knowledge? How granites behave?
 (you might just note that you discuss this in the discussions but… I think it is important to point out what forms of conceptual knowledge that geologists use are NOT in your model here…)

60: A cross-section is an artefact o f us working with a-c reasoning for a good enough (the A.I. term is satisficing) 3d model. It feels like you are stumbling over 2 fundamentally different *kinds* of constraints: this unit has "this place and this geometry" versus "this unit is *younger* than that unit."  The kinds of inference / constraints used in geological reasoning are well studied e.g. for the case of how people map and how they interpret data (e.g. Bond and students, etc. etc.). I don't expect a huge change here but flag *something* about what is in and what is not for your discussion later asper the previous comment?

63: Do you really want to reproduce them? Or get the same or better resulting outputs with reproducibility?

67: Are you capturing stuff people don't "bother" to do or stuff that is intractable to do manually at all?

70: Is the summary human readable? Further to my general comment, would that be desirable for human AI model development? Mostly addressed, but…

94: Given the point about modeler 'in the head' knowledge and 3d Models I might say 'un-related to 3d Modelling as currently practiced'? Not sure…

100-104: Was there any co-evolution of these libraries and your stuff as your project went on or are you totally arms-length?

(2)

114: I think you need to cite something on the geology of the area? A report?

118: Confusing. I read this as you saying the Jupyter notebook hits those layers and then provides a human-in-the-loop UI/Configuration file writer / process? Consider putting the "there are currently" sentence first for clarity?

122: It feels like you are saying this is a legend-language style depiction but I think you are actually saying it is a recoded _map_ in a unique chronostratigraphic order. So … someone has to 'do' that first…?

133-43: How do you / can you deal with adjacent map sheets with different levels of detail? Or are you 'one map only' or would a prelim recode be required? Perhaps in discussion or not at all.

145: Was this map chrono… suitable or did you recode?

A general question about this point. In field geology we use 'established, inferred, assumed' language for contacts. I'm assuming that you just have one contact type. This becomes VERY important in your discussion and conclusions when you talk about issues because not all contacts are fixed. Some are topologically required but can be 'safely' moved as they are under cover / underconstrained. So… is the 'fixist' logic HERE a problem later when you might want to reconcile some of your issues by shifting a boundary in the map because it is 'unbound' or 'unconstrained?'

191-3: I find this confusing. The way it is worded L167 filtered by… implies that it is done FIRST as data prep. But is it happening along the way? During the run?

It feels like you should discriminate things that are done 'totally before' form those that happen during… Yes, I'm being a bit picky here but I'm trying to understand the workflow (e.g. general comment about what the experience of using it is like).

(3)

217: Not sure this is meaningful because I'm not reading your code, but would it be possible to modify Figure 4 to show what is map2model and what is map2loop, or is the back and forth too complex to make that meaningful?

220: DTM is online only whereas other data can be local?

231: What about lenses in stratigraphic sequences that pinch out? Can you handle that topology?

255: Is the unique fault name auto-generated if the fault was not named in the source data?

325: 3.2 Need to remind us what these are, how they are stored? Sentence here feels like it should actually start 3.2.1 and not be a general section intro?

330: Overrides – implies you either would do this before, or during, or after and rerun, or??? Clarify the workflow please!

380: This is an area where I hoped in the discussion there would be more … discussion … about how you validated your tool as you worked on it? As in, more discussion of running with different map types, geometries etc. It is a bit too succinct there for my liking but perhaps you are at a length limit for the paper…

390: At this point I think a high level description would help. Something about 'first we do the stratigraphy, then the intrusions, then the faults, then the…. I struggled reading this trying to decide whether you were using stratigraphy as a catch all term or just for… stratigraphy. Your language is a bit ambiguous here, and a very short intro on the process would help to say what is covered and what is not.

414: It feels like this could be a parallel combinatorial approach; mostly addressed in the discussion.

427: is the graph exportable/ usable?

439: Again, the workflow. Is this process interactive? Iterative? Black box? Is the human in the loop? Especially given the statement below (geoscientist in minutes etc). Expose your workflow more clearly!!!

449: (perhaps) one sentence on what geophysics would add?

514: No map… Actually as you hint at later, there are MANY situations where a human can make a map using regional knowledge, stated assumptions, theory, potential field data… The interesting thing is the gap between formally decidable maps and what a geoscientist would be comfortable to handle. There is abundant literature on this e.g. in field mapping and in interpretation (Bond, as you cite). Is the gap large? What are those humans DOING? Are they nuts? Brodaric has talked a lot about this as have the cognitive scientists who study the mapping process. There is a fairly interesting philosophy of geology topic you're going to hit if you continue in this direction (as an aside, this has been examined in the case of highly constrained geotechnical 3d modeling, but only at the very local scale and for a few unconstrained parameters). **There is a huge gap between a provable and a plausible model.**

550: Fascinating topic. Could something like a generative adversarial network approach be used (build a procedural simulator, like Noddy on steroids, that then subsets it's outputs to feed into map2loop which can then be verified since … the full inputs were known).

565: This is the combinatorics I was referring to. Specifically, the work that has been done on combinatorial possible worlds (this is just an aside…)

A general comment, I was surprised that Varnes' paper doesn't show up in your early intro as it was the source of a lot of early thinking. Not a requirement, but it is an interesting read!

***Technical Corrections***

50: … and to extract …?

52: Commonly… often… two in the same sentence seems a little wishy-washy.

53: or, if available, logged well data…?

55-56: Feels like 34d category should be in same sentence? Feels like an edit error here.

67: Not previously available.  (missing .)

68: … from GIS layers stored locally or online servers?

80: One too many ) ?

81: these packages?

84: and so this necessary toolset will not be discussed further here?

361: A single item list?

366: Compare reference at 335 to that here and make your style consistent with journal standards?

389 / 834: Perrin reference is incomplete? I didn't go through them all very carefully but…

---

## Referee Comment (RC2)

This article is in an important area for geology - building models of the subsurface quickly, using all or most available data and highlighting uncertainties. The paper is an important contribution in terms of the software that it is offering for the community and demonstrating this software with a series of datasets in a case study. However, the article suffers from a number of issues that mean that I would advise the authors reorganising the paper, running some additional benchmarking and documenting how efficient the code is as well as how it works in terms of datatypes and inputs and outputs. The paper would also benefit from a better review of the literature in places, especially the introduction and discussion. The comments below are as detailed as I could be to improve the paper and should not require a lot of additional scientific work, mainly rewriting and organising the text.

**General Comments**

a. The article is poorly organised and confusing in its current state, mixing a several terms, data types, layer types, categories, inputs and outputs without clearly structuring their relationship. Parts of the writing leave a lot to be desired - there are a lot of unscientific terms and quite a lot of referencing missing. New paragraphs are not always given a line in between (or alternatively indented), making the article hard to read at points. The use of caplock text is annoying (e.g. heading 2.1, line 121)

b. A development and technical paper "usually include[s] a significant amount of evaluation against standard benchmarks, observations, and/or other model output as appropriate." I would not consider the example dataset given a significant amount of evaluation - other datasets such as NSW etc. are mentioned and it is also said that the time to build such models is very small ("The example map and associated data used in this paper took just over 3 minutes to deconstruct with map2loop and a further 4-15 minutes to build with the three target modelling engines, running on a standard laptop computer." - lines 456-7) so I'm not sure why more examples are not given. In addition, the inclusion of a very imprecise single runtime should not be considered a benchmark. The architecture of the machine needs to be mentioned as well ("standard laptop" does not suffice).

c. I suggest that the examples of data given in 5,7,9 and 10 are actual results of the model run against a particular dataset rather than just examples of inputs/outputs. Therefore, I would split the results section as a case study and describe the model separately. Even better, the results from this dataset could be used as an example and several case studies used to show different run-times etc. for benchmarking the results and the difficulties/comparisons drawn between these models discussed.

1. **Introduction**
a. The introduction reads more like an abstract (and indeed is rather similar to most of the abstract) rather than a detailed review of the literature in this area. There are not enough papers cited and the introduction seems to define new concepts rather than base those concepts clearly on previous work. The second half of the introduction (Lines 75-104) reads more like it belongs in the methodology section.

b. Line 53-54: "Unfortunately, away from basins and mines, drill-holes are often too shallow to provide constraints at the regional scale, and also often lack stratigraphic information." It would be great to specify in which context your proposed solution is targeted at and what other solutions for other regions would work (e.g. in basins). This might be in the introduction or in the discussion or both.

c. Lines 52-62: This paragraph is difficult to read and no references are given as examples of the types of data or on alternative methods of validation mentioned, e.g. via gravity and magnetic data (line 62).

2. **Input Data**

a. The introduction to this section does not let the reader know that the following 6 subsections correspond to the six input data types on Figure 1. The lack of consistency between headings of the subsections and the titles of the datatypes from Figure 1 makes this also difficult to understand. The text should additional specify what constraints each of the input data types have - it mentions which datasets were used to derive them for this example but not what other similar datasets (e.g. are there easily available WAROX or ASUD equivalent databases in other parts of the world? If so give us some examples)

b. More details could be provided about how they were generated for this case (e.g. data type 5). Why are some of the polygons mentioned in input 1 missing from input 5 (e.g. Hardey Formation)? How is the operation coded and where is that handled? Is it a preprocessing step or is it included in the current code? If it is preprocessing, how is this calculated? (e.g. array sorting?).

c. It is difficult for the reader to understand the difference between input data types and how they relate to the three categories of data mentioned in the introduction (position, gradient and topological) as well as how they map to the subsections in Section 2 on input data (e.g. 2.1 Chronostratigraphic layer)

d. Programming terms related to classes in the code (such as POLYGON) should be introduced in some form in the introduction and demarcated from the regular word polygon (which might suffice in any case because I think your programming classes are also regular shapes with 3 or more sides. Multipolygon/Polyline/Multipolyline could just be defined in the intro as well. Again, the headings of 2.2 seem to indicate layers contain a single type of shape (e.g. polyline) that is appropriate to a single datatype (e.g. a fault). It would be good to indicate in a table of some kind which types can be associate with which layers/datatypes etc.

3. **Augmented Data Outputs**

a. Each of the subsections of this section are poorly written, full of jargon and hard to understand what is being discussed or why. For example "For regional geological models[,] a high resolution topography model is usually not needed" - under what conditions is "usually" applied? The subsections miss introductory sentences that explain how this datatype fits into the broad categories of data outlined in the intro to section 3 (positional/gradient/topological) or why it exists and the intro to section three does not let us know why these particular datatypes exist either. This section also includes quite a bit on methodologies used to create the outputs - so perhaps this could be a methodology section? Or a separate section on methodology and then a section on outputs showing how generated data could be outputted from the code. Also could be clearer which datatype corresponds to map2model / map2loop by organsing them under those headings or mentioning this in the introduction (e.g. map2model can generate x,y,z while map2loop can generate x,a,b,c).

b. Lines 216-218: "This paper focuses on two libraries, map2model (C++), and map2loop (Python).

map2model performs a spatial and temporal topological analysis of the geological map, and map2loopfurther refines this analysis by including information from non-map sources, such as stratigraphic databases, acts as a wrapper for map2model and performs all other calculations."

c. I'm not sure if this belongs at the end of this section or was cut and paste from the introduction at some point. It seems out of place here and I'm not sure how it relates to either the preceding or following paragraphs.

**4. 3D Modelling of map2loop outputs**

a. Confusingly the title suggests that only map2loop should be included here, but line 439 immediately mentions map2model. This section is very brief and needs to be expanded or integrated into the discussion section or a results section created.

b. Line 447: "Of course the whole system relies on the quality of the input data" - why "of course" and how is this handled? Remind us of any quality checks and how the quality of the input data might affect results (or papers that show how this can be handled). Why is the integration of geophysics beyond the scope of the paper (see my comments in the introduction about basins - this should be tightly linked here)? Immediately, it seems that DEM might be considered a geophysical dataset already and if certain shapes can be extracted from geophysical data (polygons/polylines) then why can't they be included?

**5. Discussion**

a. This section reads like an uncertainty section. However, a lot of the discussion fails to discuss uncertainties very scientifically, with terms like "reasonably robust" (line 464) or "'reasonable' results" (line 469). The discussion fails to identify ways to handle the uncertainties in a well-defined way. Since there is "no unique solution" (line 470) and many choices are "arbitrary" (line 505), it seems impossible to quantify model errors or uncertainties. However, the workflow nature of the model generation and mapping of particular choices should make error estimation possible. I would have liked to have seen a more robust discussion of the literature in this regard. Section 5.3 mentions wrapping the workflow in a Bayesian network but it would be good to have this discussion linked back to the previous sections on different kinds/sources of uncertainty in the models and what the implications for these different kinds/sources are on the Bayesian framework.

b. The sensistivity analysis is mentioned to be essentially trivial (line 565), so why not perform it for the current dataset?

**6. Conclusions**

I'm not sure that all the conclusions are carefully based on the results of the paper. In particular:
   a. It has not been proven (only claimed) that the model reduction is significantly reduced.
   b. Priors (a Bayesian concept) is first mentioned in the conclusion
   c. And see my remarks about uncertainty above, that this provides a homogeneous pathway to sensitivity analysis etc - since these are only claimed to be in development.

**Minor Issues:**

1. Line 52: New paragraph or not?
2. Line 56: In a 3D workflow; (, should replace ;)
3. Line 56-59: Difficult to understand what is meant here:
4. *In a 3D workflow; spatial and temporal topological data, such as the age relationships between*

*faults and stratigraphic units, we combine all of these direct observations with conceptual information, based on our understanding of the tectonic history of the region, including assumptions regarding the subsurface geometry of faults and plutons, to provide sufficient constraints to build a 3D geological model.*

5. Lines 206-208*: These are the same types as input data. Why not refer readers back to the definition of these types in the introduction.

6. *These outputs are grouped by type: positional outputs, which provide information on the location and shape of features; gradient outputs, which provide information on the orientation of features; and, topological outputs that provide information on the spatial and temporal relationships between features.*

7. Line 248 and Line 258: are essentially the same: "Processing of X geometries consists essentially of extracting the x,y location…."

8. POLYGON vs polygon (etc.): An alternative is just to use "Polygon" after defining it. However, the word polygon with lower case could probably suffice in most cases. In some cases, the capslock is not used consistently, e.g. line 172 (POINTS) and 173 (point) or in heading 2.1 ("Chronostratigraphic POLYGON and MULTIPOLYGON layer") and Figure 1 ("1. chronostratigraphic polygons"). Consistency is a problem, e.g. Line 288: Is the series of x,y,z points - are these POINTS or points? Also, for example, Figure 3 references POLYGONS etc. in caption by "polylines" in the figure insert.

From <https://gmd.copernicus.org/preprints/gmd-2020-400/gmd-2020-400.pdf>

---

## Author Response (AR1)

Response to reviewer's comments RC1

We would like to thank the reviewer for their careful reading of the text and numerous suggestions, most of which we have incorporated into the manuscript. Specific responses are included below.

Line numbers after each comment refer to the line numbers of the new clean version of the manuscript.

R.M. Harrap Review

Scientific Specific Comments and Requests for Clarification

Some of these are quite minor points, some may reflect me misunderstanding the paper (which says something) and some are fairly technical. The format is line number or range followed by the point, however large or trivial.

30: Why say deformed? Doesn't it work on flat-lying strata?
The references we refer to here are specifically focused on deformed rocks, so flat-lying strata are the pre-cursor geometries but not the focus of these authors' works.

35: The point of this work (I'm the author) was consistency checking both during and after map creation and especially when large, complex compilation maps were being created and to focus on areas where a legend contradicts map relationships. This was implemented at one point in an unpublished MSc thesis and it worked. The point here is that workers such as Brodaric and Harrap were always interested in tools used during mapping as well as during compilation and not simply in analyzing others maps. Not sure this requires any change in the paper, but I thought it interesting to point out given the long and slow history of developments of tools in this area!
Interesting, I have used some of your text here to better define the original aims (from line 33 of clean version).

43: I'm not sure what irreproducible means here.
Clarified in text to refer to two geologists building models of the same area from the same data, or one geologist at different times (Line 57)

46: Is that transformation unique? Or is it an interpretation especially in cases where legends are being adapted? Brodaric, and Coleman-Sadd (iirc) did some work on reconciling maps where sub-maps had different authors and different emphasis during mapping.
Non-uniqueness highlighted here and reference to Coleman-Sadd paper added. (Line 63)

51: As well as for use in other studies?
Corrected  (Line 74)

54: … lack stratigraphic information. This reads oddly since drill holes in non-hardrock terrain would never lack stratigraphic information? Perhaps I'm misunderstanding here.
Clarified that most non-basin drillholes lack strat information  (Line 78)

55: The 3 types specified here – might be expanded with one sentence of more detail on each? And why (just) these three? As you'll see below I have comments about others though you address that very well in your Discussion in the end.
Examples for each type are now given. (from Line 102)

58: Combine direct with conceptual?
What is knowledge from nearby areas?
Tectonic history (okay, you cover this)
What about base scientific knowledge? How granites behave?
(you might just note that you discuss this in the discussions but… I think it is important to point out what forms of conceptual knowledge that geologists use are NOT in your model here…)
This section has been expanded (from Line 102).

60: A cross-section is an artefact of us working with a-c reasoning for a good enough (the A.I. term is satisficing) 3d model. It feels like you are stumbling over 2 fundamentally different kinds of constraints: this unit has "this place and this geometry" versus "this unit is younger than that unit." The kinds of inference / constraints used in geological reasoning are well studied e.g. for the case of how people map and how they interpret data (e.g. Bond and students, etc. etc.). I don't expect a huge change here but flag something about what is in and what is not for your discussion later asper the previous comment?
Added text clarifying specific and generic constraints (From Line 104)

63: Do you really want to reproduce them? Or get the same or better resulting outputs with reproducibility?
Good point, changed to inspired by… (Line 114)

67: Are you capturing stuff people don't "bother" to do or stuff that is intractable to do manually at all?
Good point, text modified to suggest that the work is not often carried out… (Line 119)

70: Is the summary human readable? Further to my general comment, would that be desirable for human AI model development? Mostly addressed, but…
Yes it is and this is explicitly stated now  (Line 124)

94: Given the point about modeler 'in the head' knowledge and 3d Models I might say 'un- related to 3d Modelling as currently practiced'? Not sure…
Clarified this statement

100-104: Was there any co-evolution of these libraries and your stuff as your project went on or are you totally arms-length?
No co-evolution, this is explicitly stated now. (Line 93)

114: I think you need to cite something on the geology of the area? A report?
Reference added  (Line 137)

118: Confusing. I read this as you saying the Jupyter notebook hits those layers and then provides a human-in-the-loop UI/Configuration file writer / process? Consider putting the "there are currently" sentence first for clarity?
A new figure 2 explaining overall workflow and showing the automated part of the system has been added. (Fig 2)
122: It feels like you are saying this is a legend-language style depiction but I think you are actually saying it is a recoded _map_ in a unique chronostratigraphic order. So … someone has to 'do' that first…?
Clarified so it is clear I am talking about polygons with stratigraphic attributes. (Line 662)

133-43: How do you / can you deal with adjacent map sheets with different levels of detail? Or are you 'one map only' or would a prelim recode be required? Perhaps in discussion or not at all.

We can't and the ref to Coleman-Sadd et al., 1997 is added to cover this possibility earlier in the section. (Line 64)

145: Was this map chrono… suitable or did you recode?

A general question about this point. In field geology we use 'established, inferred, assumed' language for contacts. I'm assuming that you just have one contact type. This becomes VERY important in your discussion and conclusions when you talk about issues because not all contacts are fixed. Some are topologically required but can be 'safely' moved as they are under cover / underconstrained. So… is the 'fixist' logic HERE a problem later when you might want to reconcile some of your issues by shifting a boundary in the map because it is 'unbound' or 'unconstrained?'
In answer to the question, the map was suitable and this point is made, and the point about inferred contacts was added to the discussion.

191-3: I find this confusing. The way it is worded L167 filtered by… implies that it is done FIRST as data prep. But is it happening along the way? During the run? It feels like you should discriminate things that are done 'totally before' form those that happen during… Yes, I'm being a bit picky here but I'm trying to understand the workflow (e.g. general comment about what the experience of using it is like).
The Figure 2 workflow clarifies this.

217: Not sure this is meaningful because I'm not reading your code, but would it be possible to modify Figure 4 to show what is map2model and what is map2loop, or is the back and forth too complex to make that meaningful?
Clarification added to figure caption.

220: DTM is online only whereas other data can be local?
This point is clarified. (Line 222)

231: What about lenses in stratigraphic sequences that pinch out? Can you handle that topology?
This point is discussed. (Line 271)

255: Is the unique fault name auto-generated if the fault was not named in the source data?
This point is clarified. (Line 297)

325: 3.2 Need to remind us what these are, how they are stored? Sentence here feels like it should actually start 3.2.1 and not be a general section intro?
Section intros added to clarify meaning of position, gradients, topology.

330: Overrides – implies you either would do this before, or during, or after and rerun, or??? Clarify the workflow please!
Workflow clarified in Figure 2.

380: This is an area where I hoped in the discussion there would be more … discussion … about how you validated your tool as you worked on it? As in, more discussion of running with different map types, geometries etc. It is a bit too succinct there for my liking but perhaps you are at a length limit for the paper…
Discussion has been expanded to include comparison with manual data extraction, choosing the data to model, different geological terranes and lack of contact information. The discussion section has been expanded and several discussion-like points have been moved from main body here as well. (from Line 540)

390: At this point I think a high level description would help. Something about 'first we do the stratigraphy, then the intrusions, then the faults, then the…. I struggled reading this trying to decide whether you were using stratigraphy as a catch all term or just for… stratigraphy. Your language is a bit ambiguous here, and a very short intro on the process would help to say what is covered and what is not.
Clarification that extraction of igneous, fault and stratigraphic contacts all take place. (Line 438)

414: It feels like this could be a parallel combinatorial approach; mostly addressed in the discussion.
Yes, a specific point is made highlighting this. (Line 425)

427: is the graph exportable/ usable?
Yes, and this is now mentioned. (Line 467)

439: Again, the workflow. Is this process interactive? Iterative? Black box? Is the human in the loop? Especially given the statement below (geoscientist in minutes etc). Expose your workflow more clearly!!!
Workflow clarified in Figure 2.

449: (perhaps) one sentence on what geophysics would add?
Done. (Line 648)

514: No map… Actually as you hint at later, there are MANY situations where a human can make a map using regional knowledge, stated assumptions, theory, potential field data… The interesting thing is the gap between formally decidable maps and what a geoscientist would be comfortable to handle. There is abundant literature on this e.g. in field mapping and in interpretation (Bond, as you cite). Is the gap large? What are those humans DOING? Are they nuts? Brodaric has talked a lot about this as have the cognitive scientists who study the mapping process. There is a fairly interesting philosophy of geology topic you're going to hit if you continue in this direction (as an aside, this has been examined in the case of highly constrained geotechnical 3d modeling, but only at the very local scale and for a few unconstrained parameters). There is a huge gap between a provable and a plausible model.
Agreed, and added to discussion in section 6.2.1.

550: Fascinating topic. Could something like a generative adversarial network approach be used (build a procedural simulator, like Noddy on steroids, that then subsets it's outputs to feed into map2loop which can then be verified since … the full inputs were known).
We have actually developed a Noddy multi-million model suite for training CNN algorithms, now referenced here. (Line 711)

565: This is the combinatorics I was referring to. Specifically, the work that has been done on combinatorial possible worlds (this is just an aside…)
No change needed

A general comment, I was surprised that Varnes' paper doesn't show up in your early intro as it was the source of a lot of early thinking. Not a requirement, but it is an interesting read!
Added to introduction (Line 31)

Technical Corrections

All technical corrections fixed

50: … and to extract …?

52: Commonly… often… two in the same sentence seems a little wishy-washy. 53: or, if available, logged well data…?

55-56: Feels like 34d category should be in same sentence? Feels like an edit error here. 67: Not previously available.  (missing .)

68: … from GIS layers stored locally or online servers? 80: One too many ) ?

81: these packages?

84: and so this necessary toolset will not be discussed further here? 361: A single item list?

366: Compare reference at 335 to that here and make your style consistent with journal standards?

389 / 834: Perrin reference is incomplete? I didn't go through them all very carefully but…

Response to reviewer's comments RC2

We would like to thank the reviewer for their careful reading of the text and numerous suggestions, most of which we have incorporated into the manuscript. Specific responses are included below.

Line numbers after each comment refer to the line numbers of the new clean version of the manuscript.

Stuart Clark Review

This article is in an important area for geology - building models of the subsurface quickly, using all or most available data and highlighting uncertainties. The paper is an important contribution in terms of the software that it is offering for the community and demonstrating this software with a series of datasets in a case study. However, the article suffers from a number of issues that mean that I would advise the authors reorganising the paper, running some additional benchmarking and documenting how efficient the code is as well as how it works in terms of datatypes and inputs and outputs. The paper would also benefit from a better review of the literature in places, especially the introduction and discussion. The comments below are as detailed as I could be to improve the paper and should not require a lot of additional scientific work, mainly rewriting and organising the text.

General Comments

The article is poorly organised and confusing in its current state, mixing a several terms, data types, layer types, categories, inputs and outputs without clearly structuring their relationship. Parts of the writing leave a lot to be desired - there are a lot of unscientific terms and quite a lot of referencing missing. New paragraphs are not always given a line in between (or alternatively indented), making the article hard to read at points. The use of caplock text is annoying (e.g. heading 2.1, line 121)
The structure has been revised with additional tables and figures to make it clearer. As it was not clear that the caplock text was referring to specific data types, we have changed them all to capitalized forms. (Throughout text)

A development and technical paper "usually include[s] a significant amount of evaluation against standard benchmarks, observations, and/or other model output as appropriate." I would not consider the example dataset given a significant amount of evaluation - other datasets such as NSW etc. are mentioned and it is also said that the time to build such models is very small ("The example map and associated data used in this paper took just over 3 minutes to deconstruct with map2loop and a further 4-15 minutes to build with the three target modelling engines, running on a standard laptop computer." - lines 456-7) so I'm not sure why more examples are not given. In addition, the inclusion of a very imprecise single runtime should not be considered a benchmark. The architecture of the machine needs to be mentioned as well ("standard laptop" does not suffice).
As there are no other codes (to our knowledge) that undertake equivalent data extraction from maps, the concept of benchmarking seems odd, however we have added a comparison with the manual extraction of the same data. We have also specified the computer used for the calculations. (From Line 540)

I suggest that the examples of data given in 5,7,9 and 10 are actual results of the model run against a particular dataset rather than just examples of inputs/outputs. Therefore, I would split the results section as a case study and describe the model separately. Even better, the results from this dataset could be used as an example and several case studies used to show different run-times etc. for benchmarking the results and the difficulties/comparisons drawn between these models discussed.
We agree with this suggestion and separated the text into Method and Results sections. (New sections 4 &

5)

Introduction
The introduction reads more like an abstract (and indeed is rather similar to most of the abstract) rather than a detailed review of the literature in this area. There are not enough papers cited and the introduction seems to define new concepts rather than base those concepts clearly on previous work. The second half of the introduction (Lines 75-104) reads more like it belongs in the methodology section. Agreed, and this has been fixed. (From Line 28)

Line 53-54: "Unfortunately, away from basins and mines, drill-holes are often too shallow to provide constraints at the regional scale, and also often lack stratigraphic information." It would be great to specify in which context your proposed solution is targeted at and what other solutions for other regions would work (e.g. in basins). This might be in the introduction or in the discussion or both.
This has been added (Line 65)

Lines 52-62: This paragraph is difficult to read and no references are given as examples of the types of data or on alternative methods of validation mentioned, e.g. via gravity and magnetic data (line 62).
References have been added and the text clarified. (Line 110)

Input Data
The introduction to this section does not let the reader know that the following 6 subsections correspond to the six input data types on Figure 1. The lack of consistency between headings of the subsections and the titles of the datatypes from Figure 1 makes this also difficult to understand. The text should additional specify what constraints each of the input data types have - it mentions which datasets were used to derive them for this example but not what other similar datasets (e.g. are there easily available WAROX or ASUD equivalent databases in other parts of the world? If so give us some examples)
The text has been revised to clarify and or build upon these points (Line 142)

More details could be provided about how they were generated for this case (e.g. data type 5). Why are some of the polygons mentioned in input 1 missing from input 5 (e.g. Hardey Formation)?
Caption clarified that this is a subset of the full extract from the ASUD. (Figure 1 caption)

How is the operation coded and where is that handled? Is it a preprocessing step or is it included in the current code? If it is preprocessing, how is this calculated? (e.g. array sorting?).
    Method is explained more clearly. (Line 205 onwards)

It is difficult for the reader to understand the difference between input data types and how they relate to the three categories of data mentioned in the introduction (position, gradient and topological) as well as how they map to the subsections in Section 2 on input data (e.g. 2.1 Chronostratigraphic layer)
This has been clarified by adding introductory text to each section.

Programming terms related to classes in the code (such as POLYGON) should be introduced in some form in the introduction and demarcated from the regular word polygon (which might suffice in any case because I think your programming classes are also regular shapes with 3 or more sides. Multipolygon/Polyline/Multipolyline could just be defined in the intro as well. Again, the headings of 2.2 seem to indicate layers contain a single type of shape (e.g. polyline) that is appropriate to a single datatype (e.g. a fault). It would be good to indicate in a table of some kind which types can be associate with which layers/datatypes etc.
Table 2 has been added to clarify geometric data types used by each feature.

Augmented Data Outputs

Each of the subsections of this section are poorly written, full of jargon and hard to understand what is being discussed or why. For example "For regional geological models[,] a high resolution topography model is usually not needed" - under what conditions is "usually" applied?
Clarified (Line 260)

The subsections miss introductory sentences that explain how this datatype fits into the broad categories of data outlined in the intro to section 3 (positional/gradient/topological) or why it exists and the intro to section three does not let us know why these particular datatypes exist either. This section also includes quite a bit on methodologies used to create the outputs - so perhaps this could be a methodology section? Or a separate section on methodology and then a section on outputs showing how generated data could be outputted from the code. Also could be clearer which datatype corresponds to map2model / map2loop by organsing them under those headings or mentioning this in the introduction (e.g. map2model can generate x,y,z while map2loop can generate x,a,b,c).
A new flow diagram explaining the overall program flow has been added (Figure 2) and introductions to each section have been added.

Lines 216-218: "This paper focuses on two libraries, map2model (C++), and map2loop (Python).

map2model performs a spatial and temporal topological analysis of the geological map, and map2loopfurther refines this analysis by including information from non-map sources, such as stratigraphic databases, acts as a wrapper for map2model and performs all other calculations."

I'm not sure if this belongs at the end of this section or was cut and paste from the introduction at some point. It seems out of place here and I'm not sure how it relates to either the preceding or following paragraphs.
Text has been edited and moved to correct location in introduction to input data section. (Line 240)

3D Modelling of map2loop outputs

Confusingly the title suggests that only map2loop should be included here, but line 439 immediately mentions map2model. This section is very brief and needs to be expanded or integrated into the discussion section or a results section created.
Section title have been changed, and sections restructured for clarity.

Line 447: "Of course the whole system relies on the quality of the input data" - why "of course" and how is this handled?
Remind us of any quality checks and how the quality of the input data might affect results (or papers that show how this can be handled). Why is the integration of geophysics beyond the scope of the paper (see my comments in the introduction about basins - this should be tightly linked here)?
 Additional text added to clarify this point in the discussion

Immediately, it seems that DEM might be considered a geophysical dataset already and if certain shapes can be extracted from geophysical data (polygons/polylines) then why can't they be included?
Additional text added to discuss use of geophysics in this context.

Discussion

This section reads like an uncertainty section. However, a lot of the discussion fails to discuss uncertainties very scientifically, with terms like "reasonably robust" (line 464) or "'reasonable' results" (line 469). The

discussion fails to identify ways to handle the uncertainties in a well- defined way. Since there is "no unique solution" (line 470) and many choices are "arbitrary" (line 505), it seems impossible to quantify model errors or uncertainties. However, the workflow nature of the model generation and mapping of particular choices should make error estimation possible. I would have liked to have seen a more robust discussion of the literature in this regard.

A paragraph refereeing to previous uncertainty work in the 3D domain has been added to Section 6.3.

Section 5.3 mentions wrapping the workflow in a Bayesian network but it would be good to have this discussion linked back to the previous sections on different kinds/sources of uncertainty in the models and what the implications for these different kinds/sources are on the Bayesian framework.

The sensistivity analysis is mentioned to be essentially trivial (line 565), so why not perform it for the current dataset?

This is an ongoing study by our group so will be treated in a separate paper.

Conclusions

I'm not sure that all the conclusions are carefully based on the results of the paper. In particular:

It has not been proven (only claimed) that the model reduction is significantly reduced.

A comparison with manual data extraction has been added to the first part of the Discussion

Priors (a Bayesian concept) is first mentioned in the conclusion

The explicit mention of conceptual priors is added to the discussion in section 6.2.4

And see my remarks about uncertainty above, that this provides a homogeneous pathway to sensitivity analysis etc - since these are only claimed to be in development.

This assertion is expanded upon in the first part of the discussion section.

Minor Issues:

All minor issues have been fixed, except where noted

Line 52: New paragraph or not?
        Line 56: In a 3D workflow; (, should replace ;)
Line 56-59: Difficult to understand what is meant here:
In a 3D workflow; spatial and temporal topological data, such as the age relationships between faults and stratigraphic units, we combine all of these direct observations with conceptual information, based on our understanding of the tectonic history of the region, including assumptions regarding the subsurface geometry of faults and plutons, to provide sufficient constraints to build a 3D geological model.
Lines 206-208: These are the same types as input data. Why not refer readers back to the definition of these types in the introduction.
These outputs are grouped by type: positional outputs, which provide information  on  the location and shape of features; gradient outputs, which provide information on the orientation of features; and, topological outputs that provide information on the spatial and temporal relationships  between features.
Line 248 and Line 258: are essentially the same: "Processing of X geometries consists essentially of extracting the x,y location….".

We have left this as is as we believe it makes things clearer for the reader.

POLYGON vs polygon (etc.): An alternative is just to use "Polygon" after defining it. However, the word polygon with lower case could probably suffice in most cases. In some cases, the capslock is not used

consistently, e.g. line 172 (POINTS) and 173 (point) or in heading 2.1 ("Chronostratigraphic POLYGON and MULTIPOLYGON layer") and Figure 1 ("1. chronostratigraphic polygons"). Consistency is a problem, e.g. Line 288: Is the series of x,y,z points - are these POINTS or points? Also, for example, Figure 3 references POLYGONS etc. in caption by "polylines" in the figure insert.

---

## Referee Report (RR1)

Harrap – Edits Round 2 – Jessell et al 2021

56-59 sentence is broken

67 missing .

80  the sentence implies you also provide everything these tools would possibly hope for…
implied anyways (just being picky).

84 yes and no. If you had just a section, fine. If you had a well constrained section and a map,
then there is obviously more information there in 3d … intro structural geology labs… ☺

144-145. Already stated. Unless you are adding particulars e.g. as you do at 157 you can
probably delete.

196 reads a bit oddly. Are you saying numbe of servers because you cited one of many servers
or am I missing something? Perhaps reword?

235-240 is this coded directly as … code … or do you use a rule base or rule table system for
later extensibility (just curious, probably doesn't need to be added, but if you did, that's
interesting.

315-324  what about growth faults (which you discuss much later in another contxt)

326 first , is unnecessary.

431 improper use of : use ( ) instead

440 say or not and?

455 have not can?

470 Jessell et al 2014 is not in the reference list. Careful, you might offend the guy.

562 At several points in the paper you mentioned things that are underway as studies. It feels
like these should be restated here. Some are not.

580  I would argue that it eventually provides a testbed for other kinds of studies, for example
in education, looking at cost effectiveness of drilling, etc. etc.  Also, if this ever worked robustly
and easily for the average geologist I could imagine using it iteratively while doing mapping.
Though in that case one might just build a proper 3d system rather than doing multi-tool with
extraction…

---

## Author Response (AR2)

**Stuart Clarke**

Suggestions for revision or reasons for rejection (will be published if the paper is accepted for final publication)

1.  I would like the following to be clarified for the readers – adding a component on how these examples might be run (e.g. from which databases the data can be obtained, which components would be needed) – this would be very helpful for potential students wanting to use map2loop/map2model for these regions.

Line 560: The choices made by the map2loop and map2model code are inspired by the thought processes of geologists when manually building a 3D geological model from the same data. There are many small or large decisions and assumptions that are made when developing the model, and the discussion below highlights some of the areas where further work needs to be done to reproduce the manual workflow. In this paper we have used an example from Western Australia, however similar examples for the Northern Territories, New South Wales, Victoria, Queensland, Tasmania and South Australia can be run using the map2loop library).

This has been added at line 563

2.  Others have worked on automated extraction of data from geophysical data, e.g automated fault extraction Wu & Hale (2016) [https://library.seg.org/doi/abs/10.1190/geo2015-0380.1] and stratigraphic units Bugge et al. (2020) [https://library.seg.org/doi/abs/10.1190/geo2019-0413.1] that could be mentioned here.

Reference to these works has been added

Line 649: al., 2021), and is beyond the scope of this paper, but could help to define subsurface orientations or even the (automatic?) extraction of geological structures from geophysical data

Reference to Wu & Hale added here to cover this point.

**Harrap – Edits Round 2 – Jessell et al 2021**

56-59 sentence is broken

Fixed

67 missing .

fixed

80 the sentence implies you also provide everything these tools would possibly hope for… implied anyways (just being picky).

Left as is

84 yes and no. If you had just a section, fine. If you had a well constrained section and a map, then there is obviously more information there in 3d … intro structural geology labs… J

Well constrained sections implies lots of drillholes or seismic in basins, otherwise it is still based on surface info. This is already discussed so left as is.

144-145. Already stated. Unless you are adding particulars e.g. as you do at 157 you can probably delete.

Deleted

196 reads a bit oddly. Are you saying numbe of servers because you cited one of many servers or am I missing something? Perhaps reword?

clarified

235-240 is this coded directly as … code … or do you use a rule base or rule table system for later extensibility (just curious, probably doesn't need to be added, but if you did, that's interesting.

Clarified

315-324 what about growth faults (which you discuss much later in another context)

Clarified

326 first , is unnecessary.

fixed

431 improper use of : use ( ) instead

Fixed

440 say or not and?

fixed

455 have not can?

fixed

470 Jessell et al 2014 is not in the reference list. Careful, you might offend the guy.

Yes it is, line 1008

562 At several points in the paper you mentioned things that are underway as studies. It feels like these should be restated here. Some are not.

Text added to summarise these efforts

580 I would argue that it eventually provides a testbed for other kinds of studies, for example in education, looking at cost effectiveness of drilling, etc. etc. Also, if this ever worked robustly and easily for the average geologist I could imagine using it iteratively while doing mapping. Though in that case one might just build a proper 3d system rather than doing multi-tool with extraction…

Left as is, as much of this is already covered

---

## Author Response (AR3)

all comments addressed in previous revision